# Contribution of the eye and of *opn4xa* function to circadian photoentrainment in the diurnal zebrafish

**Clair Chaigne[1], Dora Sapède[1,2], Xavier Cousin[3], Laurent Sanchou[4], Patrick Blader[1], Elise Cau[1] ***

**1** Unité de Biologie Moléculaire, Cellulaire et du Développement (MCD, UMR5077) Centre de Biologie Intégrative (CBI, FR 3743), Université de Toulouse 3/UPS, CNRS, UPS, Toulouse, France, **2** IRMB, Université de Montpellier, INSERM, Montpellier, France, **3** MARBEC, Université de Montpellier, CNRS, Ifremer, IRD, INRAE, Route de Maguelone, Palavas, France, **4** Centre de Biologie Intégrative (CBI, FR 3743), Université de Toulouse 3/UPS, CNRS, UPS, Toulouse, France

* elise.cau@univ-tlse3.fr

**Data Availability Statement:** All data are in the manuscript and/or supporting information files.

**Funding:** This work was supported by the Centre National de la Recherche Scientifique (CNRS);the

## Abstract

The eye is instrumental for controlling circadian rhythms in mice and human. Here, we address the conservation of this function in the zebrafish, a diurnal vertebrate. Using *lakritz (lak)* mutant larvae, which lack retinal ganglion cells (RGCs), we show that while a functional eye contributes to masking, it is largely dispensable for the establishment of circadian rhythms of locomotor activity. Furthermore, the eye is dispensable for the induction of a phase delay following a pulse of white light at CT 16 but contributes to the induction of a phase advance upon a pulse of white light at CT21. Melanopsin photopigments are important mediators of photoentrainment, as shown in nocturnal mammals. One of the zebrafish melanopsin genes, *opn4xa*, is expressed in RGCs but also in photosensitive projection neurons in the pineal gland. Pineal *opn4xa*+ projection neurons function in a LIGHT ON manner in contrast to other projection neurons which function in a LIGHT OFF mode. We generated an *opn4xa* mutant in which the pineal LIGHT ON response is impaired. This mutation has no effect on masking and circadian rhythms of locomotor activity, or for the induction of phase shifts, but slightly modifies period length when larvae are subjected to constant light. Finally, analysis of *opn4xa*;*lak* double mutant larvae did not reveal redundancy between the function of the eye and *opn4xa* in the pineal for the control of phase shifts after light pulses. Our results support the idea that the eye is not the sole mediator of light influences on circadian rhythms of locomotor activity and highlight differences in the circadian system and photoentrainment of behaviour between different animal models.

## Author summary

Experiments performed in mice have established a crucial role for the eye in general and melanopsin expressing cells in particular in the control of circadian rhythms most notably during photoentrainment, by which circadian rhythms adapt to a changing light

Institut National de la Santé et de la Recherche
Médicale (INSERM) to P.B; Université de Toulouse
III (UPS) to P.B; Fondation pour la Recherche
Médicale (FRM, https://www.frm.org/en, grant
number DEQ20131029166) to P.B; Fédération
pour la Recherche sur le Cerveau (FRC, https://
www.frcneurodon.org) to P.B; Association pour la
Recherche sur le Cancer (ARC, https://www.
fondation-arc.org) to P.B; D.S was a recipient of a
fellowship from the Association Rétina France and
C.C received a fellowship from the Ministère de la
Recherche. The funders had no role in study
design, data collection and analysis, decision to
publish, or preparation of the manuscript.

**Competing interests:** The authors have declared
that no competing interests exist.

environment. In marked contrast to this, we show that in zebrafish the eye and photosensitivity dependent on one of the melanopsin genes, *opn4xa*, which is expressed in both the eye and the pineal gland, are largely dispensable for correct circadian rhythms. These results provide insight that the light sensors orchestrating circadian rhythms of locomotor activity are different between animal models revealing that vertebrates employ different molecular/cellular circuits for photoentrainment of behavior depending on their phylogeny and/or temporal niche.

## Introduction

Light has a profound influence on the physiology and behaviour of living organisms. In particular, it controls circadian rhythms that in turn regulate a variety of biological functions. Circadian rhythms are defined by their period of approximately 24 hours. Once established, these rhythms persist in constant conditions, which has fostered the concept of an endogenous time-keeping mechanism known as the circadian system. Nonetheless, external cues are required to synchronize (or 'entrain') circadian rhythms with the exogenous environmental conditions. For instance, light entrains the circadian system through a process referred to as photoentrainment (see [1] for a review).

In mouse, photoentrainment depends on a functional retina. Eye enucleated mice or mice lacking retinal ganglion cells (RGCs) do not entrain to LD (Light/Dark) cycles and thus behave as if they were in constant darkness [2–4]. Similarly, in human, fifty percent of blind people exhibit circadian misalignment with the LD cycles [5–7]. In mouse, photoentrainment depends on a specific subtype of RGCs expressing the photopigment melanopsin, which is encoded by the *Opn4* gene. These RGCs are sensitive to blue light and are referred to as ipRGCs for "intrinsically photosensitive RGCs". Mice mutant for *Opn4* shows a diminished phase-delay in response to a pulse of light administered at circadian time 16 (CT 16; at the beginning of the subjective night) but entrain normally to LD cycles [8,9]. In contrast, mice with no ipRGCs or with impaired neurotransmission from ipRGCs show no entrainment to LD as well as no phase delay following a light pulse at CT16 [10–12]. The difference between the phenotypes observed when only melanopsin photosensitivity is impaired compared to the models where ipRGCs inputs to the brain are lost is thought to result from the influence of classical rods and cone photoreceptors on ipRGCs. Indeed, both rods and cones have been shown to play a role during photoentrainment and to signal to ipRGCs [13–19]. Thus, ipRGCs function as a hub that integrates and transmits light information to the brain through a direct projection to the core of the suprachiasmatic nucleus (SCN; [20–22]).This hypothalamic nucleus functions as a 'master clock' that synchronizes peripheral clocks present everywhere in the body.

In addition to photoentrainment, ipRGCs also control the increase of period length when animals are placed in constant light (LL) [8,9] and are required for a process of maturation of the circadian clock that sets the definitive period of locomotor rhythms in LD and in constant darkness (DD, [23]. Finally, in addition to their crucial influence on the circadian system, murine ipRGCs also control masking, a direct suppressive effect of light on locomotor activity. This activity is thought to involve different ipRGC subtypes than the ones that impact the circadian system [24].

ipRGCs are well established to mediate circadian and direct effects of light on behaviour in nocturnal mammals. Based on the conservation of the nervous system, observations made in human blind people and the description of ipRGC populations in the diurnal rodent

Arvicanthis ansorgei as well as in human, ipRGCs are considered to play a similar role in diurnal mammals [25,26]. While the zebrafish has emerged as a powerful non-mammalian diurnal vertebrate model for chronobiology, the neural circuit controlling photoentrainment of behaviors has not been identified in this species. The observation that two species of cavefishes bearing eye degeneration: *Astyanax mexicanus* and *Phreatichthys andruzzii* do not show robust light-entrainable circadian locomotor activity rhythms could suggest that eye function is crucial for photoentrainment of locomotor rhythms in fishes [27,28]. However, the observed circadian locomotor phenotypes could result from additional photoreceptive deficits. Indeed, *Phreatichthys andruzzii* present mutations in several opsin genes that are expressed in extraocular locations [28]. While the function of the eye regarding circadian rhythms of behaviors has not been addressed in fishes, in the zebrafish model, an additional level of complexity arises from the observation that all cell types are photosensitive [29–31]. This local photodetection could serve local functions (metabolism, transcription, cell cycle) or participate to the control of behaviors, although some level of central control is expected for orchestrating a complex process such as behavior. As such, the relative importance of central versus peripheral control for the photoentrainment of locomotor activity is still unclear. Finally given the widespread photosensitivity of the larval zebrafish, a central control could be exerted by a photosensitive brain nucleus in this species.

To begin addressing how photoentrainment is controlled in zebrafish we first tested the role of the retina using *lakritz* (*lak*) mutants in which all RGCs fail to develop and as such no connection exist between the eye and the CNS [32]. *lak* mutant larvae entrain to LD cycles and maintain rhythms of locomotor activity with a period similar to their control siblings in constant darkness (DD) and constant light (LL). While we detected no defect in phase shifting in response to a pulse of white light produced in the early subjective night (CT16) in *lak* -/- larvae, we observed a reduction of the phase shift induced upon a similar pulse of light at CT21 in *lak* mutants. Interestingly, the induction of the clock gene *cry1a* upon such a pulse of light is impaired in *lak* larvae compared to control siblings. The zebrafish possesses five melanopsin genes that are all expressed in the retina, including *opn4xa* and *opn4b* in larval RGCs [33,34]. In addition, melanopsin expression is detected in extra-retinal tissues. For instance, *opn4xa* is expressed in a subpopulation of projection neurons in the pineal gland. Interestingly these *opn4xa+* projection neurons function in a LIGHT ON fashion while *opn4xa-* projection neurons function in a LIGHT OFF manner [35]. We engineered an *opn4xa* mutant in which the LIGHT ON response of the pineal is impaired; *opn4xa* -/- larvae successfully entrain to LD cycles and maintain rhythms of locomotor activity in constant conditions albeit with a reduction of period in LL. Pulses of white light at CT16 and CT21 induced similar phase shifts in *opn4xa* mutant and *opn4xa/lak* double mutant larvae compared to controls. Our results suggest that the function of the retina and the LIGHT ON response of the pineal gland are not absolutely required for circadian photoentrainment in zebrafish, thus further highlighting differences in the circadian system and circadian photoentrainment between mammals and zebrafish.

## Results

### The zebrafish eye is dispensable for the establishment of circadian rhythms

Photoentrainment in mouse and human requires a functional eye. We took advantage of *lak* mutant larvae that lack RGCs to address whether this role for the eye is conserved in the diurnal zebrafish. Homozygous *lak* mutant larvae lack neuronal connections between the eye and the brain and do not display an optomotor response [32,36,37]. The gene mutated in *lak*, encoding for the bHLH transcription factor ATOH7 (ATH5) is expressed exclusively in the

developing retina [38]. The *lak*<sup>*th241*</sup> allele we used bears a point mutation and functions as a null allele which results in a total absence of both RGCs and a neural connection between the eye and the brain [32]. We compared the locomotor activity of homozygous *lak* mutants with siblings (*lak*+/+ and *lak*+/-) in different illumination conditions. For each of these conditions, three independent experiments were performed. Within each independent experiment, the same number of homozygous mutants and control siblings were randomly selected and the mean of the three experiments was plotted. In this manner, the weight of each experiment within the final mean was identical between mutant and sibling populations.

In cycles of 14h light: 10h dark (hereafter referred to as LD), both siblings and *lak* homozygous mutant larvae exhibit rhythms of locomotor activity that are aligned with the LD cycles (Fig 1B). However, compared to sibling larvae, *lak* mutants show a specific reduction of activity during the day (Fig 1B and S1 Table). Sibling and *lak* mutant LD-entrained larvae placed in constant darkness (DD) demonstrate rhythms of locomotor activity with similar levels (Fig 1C and S2 and S3 Tables). In addition, the periods of the rhythms observed in DD did not significantly differ between the two populations (Fig 1D). The reduction in activity observed during the day in LD conditions thus does not affect the persistence of rhythmicity under free running conditions in DD.

Finally, in constant light (LL) conditions, the activity of siblings and *lak* larvae were similar (Fig 1E, 1F). The period of the *lak* mutant rhythm was slightly higher than the sibling rhythm but this difference was not significant (siblings: 24.93 ± 1.87 hours (n = 66), *lak*: 25.63± 2.1 hours (n = 68); mean ± S.D; p = 0.083; Mann-Whitney). Altogether these results suggest that retinal ganglion cells and therefore a neuronal connection between the eye and the brain are dispensable for the establishment of circadian rhythms, their correct alignment to LD cycles and their maintenance in free running conditions (DD or LL).

## Absence of a functional eye differentially affects the induction of a phase delay and a phase-advance following pulses of light during the subjective night

To evaluate the role for RGCs in circadian photoentrainment, we assessed the phase-shifting effect of a pulse of white light on locomotor activity in *lak* larvae during the subjective night. We first chose to perform such a light pulse at CT16, as this was previously shown to induce a robust phase shift of the molecular clock in cell cultures [30,39]. After entraining for 5 LD cycles, larvae from *lak*+/- incrosses were further reared in DD and subjected to a pulse of light during the second night in DD ("PD larvae"). Their activity was compared with the activity of larvae kept in the dark for 4 days ("DD larvae"). To analyze if a phase shift was induced, we calculated the difference of phase between the two last days ("after the light pulse") and the two first days ("before the light pulse"), a value we refer to as "Δphase" (Fig 2A). We found that a 2-hours pulse of light at CT16 induced a phase delay of locomotor activity rhythms in larvae, as the Δphase of PD larvae was higher than the one of DD larvae (Fig 2B and Table 1). When the difference between the Δphase of pulsed larvae minus the Δphase of larvae placed in DD was calculated, it suggests a phase delay of 2.94 hours on average in PD larvae. Finally, we determined that control and *lak* mutant larvae exhibit a similar phase shift in locomotor activity (Fig 2C and Table 1) suggesting that RGCs are not necessary for the circadian photoentrainment of locomotor activity to a pulse of light at the beginning of the subjective night (CT16).

We next performed a pulse of light at the end of the subjective night. A one- hour pulse induced a phase advance that could be detected during the second circadian cycle after the pulse (Fig 2D and 2E). Since this phase shift is not seen during the first cycle, we analyzed

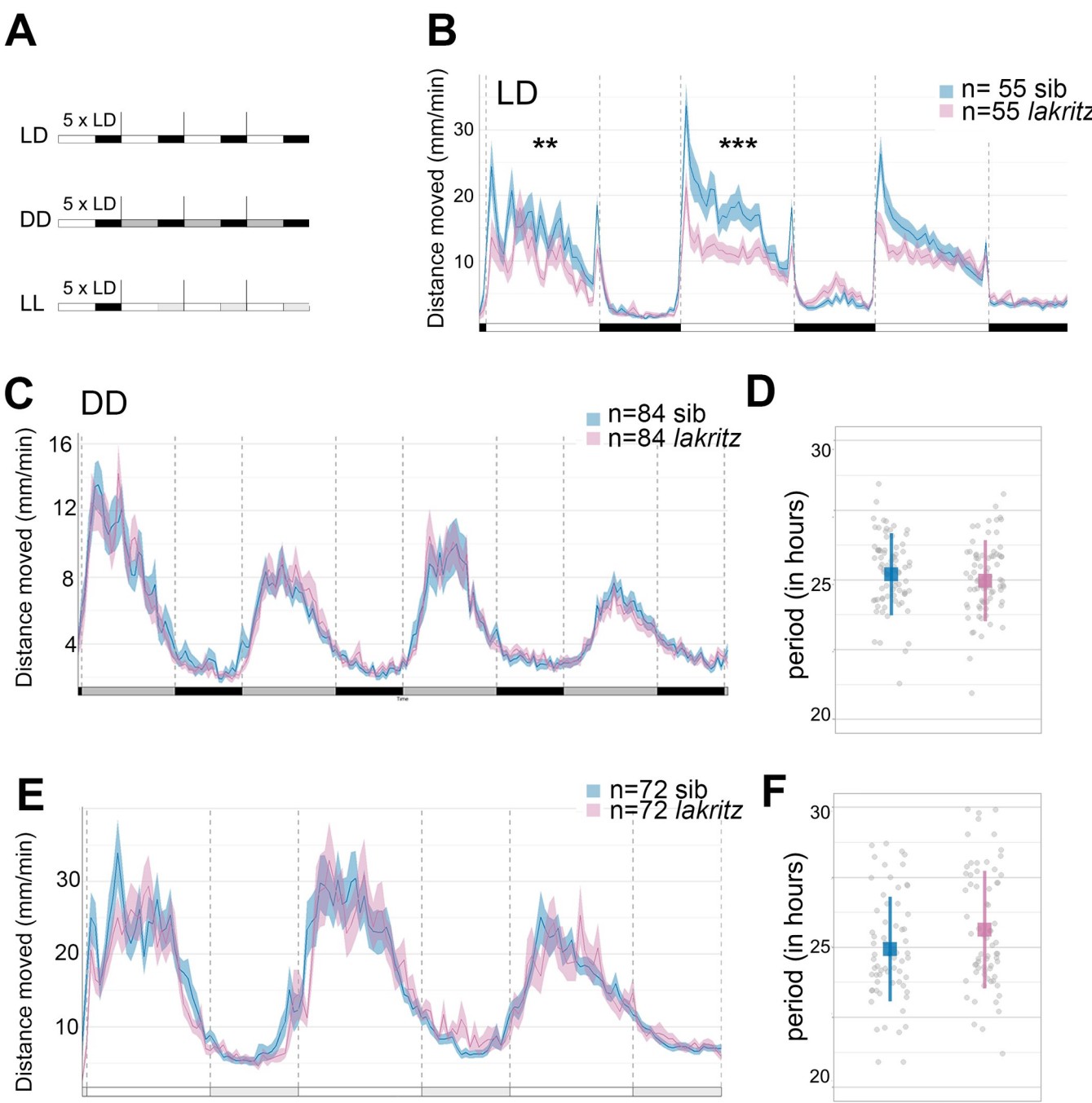

**Fig 1. Locomotor activity of larvae devoid of RGCs in LD, DD and LL. A)** Experimental design of LD, DD and LL experiments. White rectangles represent the day period, while black rectangles represent the night period, light grey rectangles represent the subjective day period and dark grey rectangles the subjective night. For each experiment, larvae are entrained for 5 LD cycles and their locomotor activity is tracked either in LD (LD), constant darkness (DD) or constant light (LL) the larvae are therefore 5dpf at the beginning of locomotor activity measurements. **B)** Average distance moved (mm/min) In LD. Merged data from 3 independent experiments represented in 10 min bins. Error bars represent SE. The distance moved is lower in *lak* larvae than control sibling larvae during the 1st (p = 0.008) and 2nd days (p = 0.005) but not during the 3rd day (p = 0.13) nor during the night (p = 0.42, p = 0.51 and p = 0.57 for the 1st, 2nd and 3rd nights. The lack of a phenotype of *lak* larvae in the third light phase could be linked to the overall state of the larvae as they are not fed during the experiment; Mann-Whitney two-tailed test), see S1 Table. * P<0.05, **p<0.01, ***p<0.005. **C)** Average distance moved in DD. Merged data from 3 independent experiments represented in 10 min bins. Error bars represent SE. No differences are detected between the distance moved of siblings versus *lak* larvae using a Mann-Whitney two-tailed test for each subjective night or day, see S2 Table. **D)** Estimation of the periods using the FFT-NLLS method. Calculations were made on four complete cycles in DD. The mean period is not significantly different between sibling and *lak* larvae in DD (control: 25.08 ± 1.59 hours (n = 114), *lak*: 24.95 ± 1.44 hours (n = 83); mean ± S.D; p = 0.32; Mann-Whitney two-tailed test, sibling vs *lak* larvae). Each grey point represents a single larva. **E)** Average distance moved in LL. Merged data from 3 independent experiments represented in 10 min bins. Error bars represent SE.

**F)** Estimation of the periods using the FFT-NLLS method. Calculations were made on three complete cycles in LL. Mean± sd (in hours) is represented. Each grey point represents a larva, n = 66 siblings, n = 68 *lak* larvae.

larvae over an additional cycle in order to obtain enough data to perform a robust phase calculation. The difference between the Δphase of pulsed larvae (PA) minus the Δphase of larvae placed in DD alone suggests that this paradigm induced a phase advance of at least 3.7 (1.8 +1.9) hours on average. While *lak -/-* larvae showed a phase advance upon a pulse of light at CT21 (Fig 2F and Table 1), this phase shift was weaker than that induced in control larvae (Δphase = -1.8 ± 7.84 for siblings versus Δphase = -0.56 ± 4.49 for *lak -/-* larvae). Although this difference in phase shift between *lak* and siblings is difficult to observe on the graph

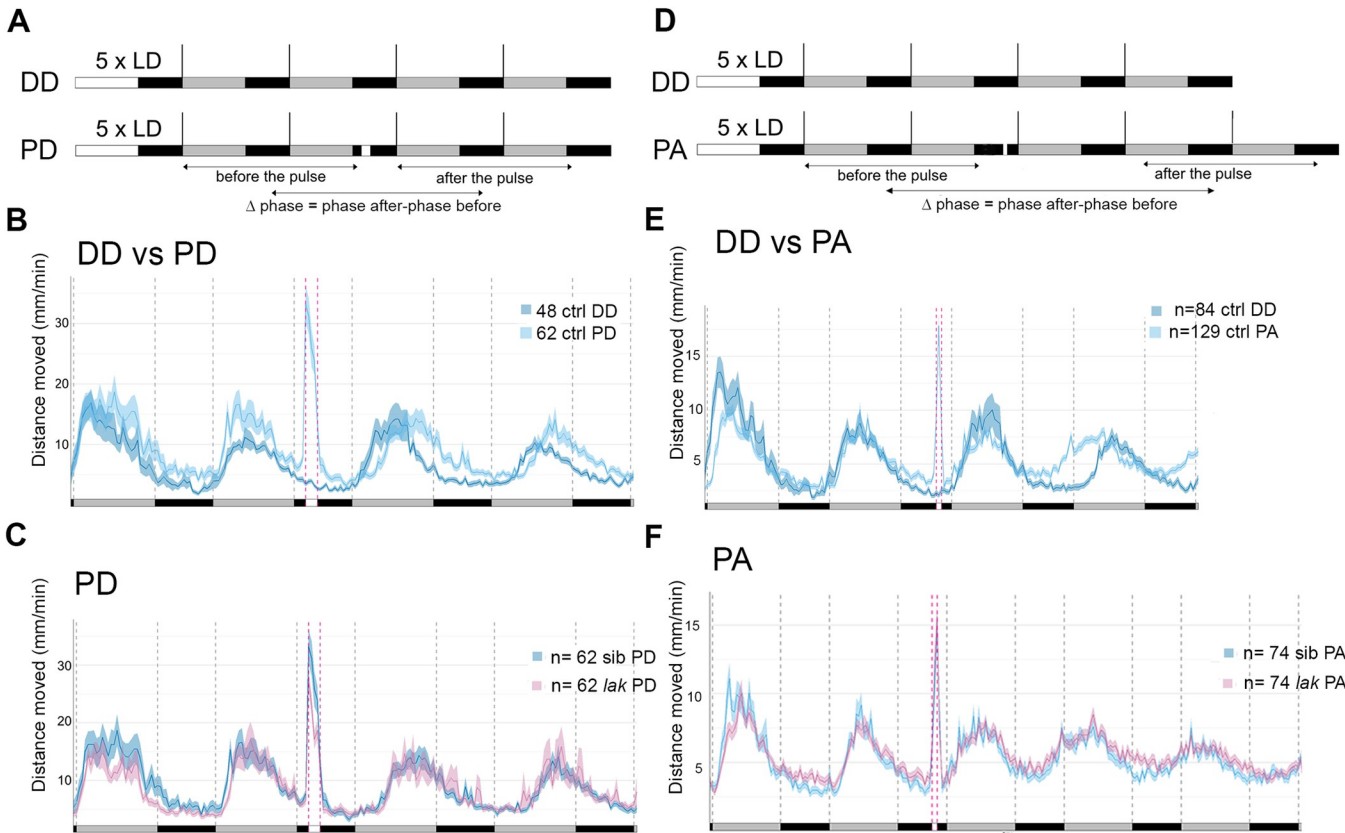

**Fig 2. Larvae devoid of RGCs still photoentrain to pulses of white light at CT16 and CT21. A)** Experimental design of phase delay (PD) experiments. White rectangles represent the day or light pulse period, black rectangles represent the night period and dark grey rectangles represent the subjective day. For each experiment, larvae are entrained for 5 LD cycles, the larvae are therefore 5dpf at the beginning of locomotor activity measurements. Locomotor activity is tracked either in constant darkness for 4 days (DD) or tracked in constant darkness for 4 days and subjected to a 2-hours pulse of light during the night of the 2nd day of constant darkness at CT16 (PD). The phase of locomotor activity is calculated for each larva before and after the timing of the pulse for DD and PD experiments and the Δphase (phase after the pulse–phase before the pulse) is calculated. **B)** Average distance moved by control larvae (mm/min over 10min) in DD and PD experiments. Mean ± SE. The Δphase calculated using the FFT-NLLS method of PD larvae is higher than the one of DD larvae (p<0.0001, Mann-Whitney two-tailed test), showing that the pulse of light induced a phase delay. **C)** Average distance moved merged from PD experiments represented in 10 min bins. Mean ± SE. The Δphase of control versus *lak* larvae calculated using the FFT-NLLS method is not significantly different (p = 0.24, Mann-Whitney two-tailed test). *lak* show lower levels of activity during the light pulse (p = 0.03, Mann-Whitney two-tailed test). **D)** Experimental design of phase advance (PA) experiments. The iconography is similar to A). PA-pulsed larvae were subjected to a one-hour pulse of light at CT21. **E)** Average distance moved of control larvae (mm/min over 10min) in DD and PD experiments Mean ± SE. The Δphase calculated using the FFT-NLLS method is negative in PA-pulsed larvae and statistically different from DD larvae (p<0.0001, Mann-Whitney two-tailed test), showing that the pulse of light induced a phase advance. **F)** Average distance moved merged from PA experiments represented in 10 min bins. Mean ± SE.

**Table 1. Quantification of the phase shifts in control siblings versus *lak*-/- (*lak*) larvae kept in DD or submitted to pulses of white light at CT16 or CT21.** The Δphase is the difference between the phase of the two last cycles and the phase of the two first cycles. A phase shift is observed in DD owing to the period that is close to 25 hours which generates a ~1 hour-shift every cycle. Upon a pulse of light at CT16 or CT21 a statistical difference is observed between DD and pulsed sib larvae as well as *lak* and sib larvae when the pulse of light is applied at CT21 (****, p<0,0001; *, p<0,05 using a Mann-Whitney two-tailed test). For each type of paradigm, (DD, PD and PA) three independent experiments were pooled.

| Condition | Δphase Mean±S.D (n) | P value Mann-Whitney two-tailed test |
|---|---|---|
| DD sib | 1.9±2.56 (73) | |
| PD-pulsed sib | 4.84±1.90 (46) | Sib: PD vs DD: **** |
| PD-pulsed *lak* | 5.72±2.32 (37) | PD *lak* vs sib: 0.24 |
| PA-pulsed sib | -1.83±7.84 (86) | Sib: PA vs DD: **** |
| PA-pulsed *lak* | -0.56±4.49 (29) | PA *lak* vs sib: * |

representing all the animals, it is more apparent when comparing only the records of animals for which a phase was successfully extracted before and after the pulse (Figs 2F and S1). These results suggest that a phase advance can occur in absence of RGCs although the eye contributes to photoentrainment in this context. Altogether, our results suggest that although phase advances and delays can occur in absence of RGCs, the absence of these cells specifically affect the response in a phase advance paradigm.

To analyse a role for RGCs during masking we calculated the activity of control and *lak* larvae during the pulses of light performed at CT16 and CT21. Interestingly, *lak* larvae showed a reduced level of activity compared to control larvae during the pulse performed at CT16 but not at CT21 (CT16: Fig 2C, siblings: 28.33 ± 19.26 mm/min over 10min (n = 62), *lak*: 21.45 ± 12.27 mm/min over 10min (n = 51); p = 0.03; Mann-Whitney two-tailed test; CT21: Fig 2F siblings: 16.51 ± 10.9 mm/min over 10min (n = 51), *lak*: 17.17 ± 7.85 mm/min over 10min (n = 62); p = 0.75). These results showed that RGCs are involved in masking in the zebrafish larvae but in a circadian dependent manner.

## *lak* mutation affects the induction of *cry1a* following a pulse of white light at CT21

To decipher the molecular mechanism underlying the reduction of a phase advance induced by a pulse of white light at CT21 in *lak* larvae, we looked at the expression of *per2* and *cry1a* in pulsed and control ('Dark') embryos at CT22. Indeed, these two genes are strongly induced by light and thought to participate in the molecular mechanism of photoentrainment [40–43]. Control embryos show expression of *per2* in the telencephalon independently of the administration of a light pulse (Fig 3A and 3B) and *lak* larvae show a similar expression compared to control (Fig 3C and 3D). In control and pulsed embryos, three different patterns of *cry1a* can be distinguished (Fig 3E, 3F and 3G): some larvae show no expression (Fig 3E), some exhibit a mild expression in the tectum and expression in the pineal (Fig 3F) while other larvae show a strong expression in the tectum, the eye, the telencephalon, habenulae and pineal (Fig 3G). In absence of a light pulse, both control and *lak* larvae show an absence of detectable expression of *cry1a*, or more rarely a mild expression. In contrast the administration of a light pulse leads to 100% larvae exhibiting a high level of expression, suggesting that *cry1a* is induced by a light pulse at CT21. This induction is much less important in *lak* larvae with only 14.3% larvae showing a strong expression pattern after a pulse of light.

Altogether our results suggest that *cry1a* and not *per2* is induced after a pulse of light at CT21 and that this *cry1a* induction is impaired in *lak* larvae.

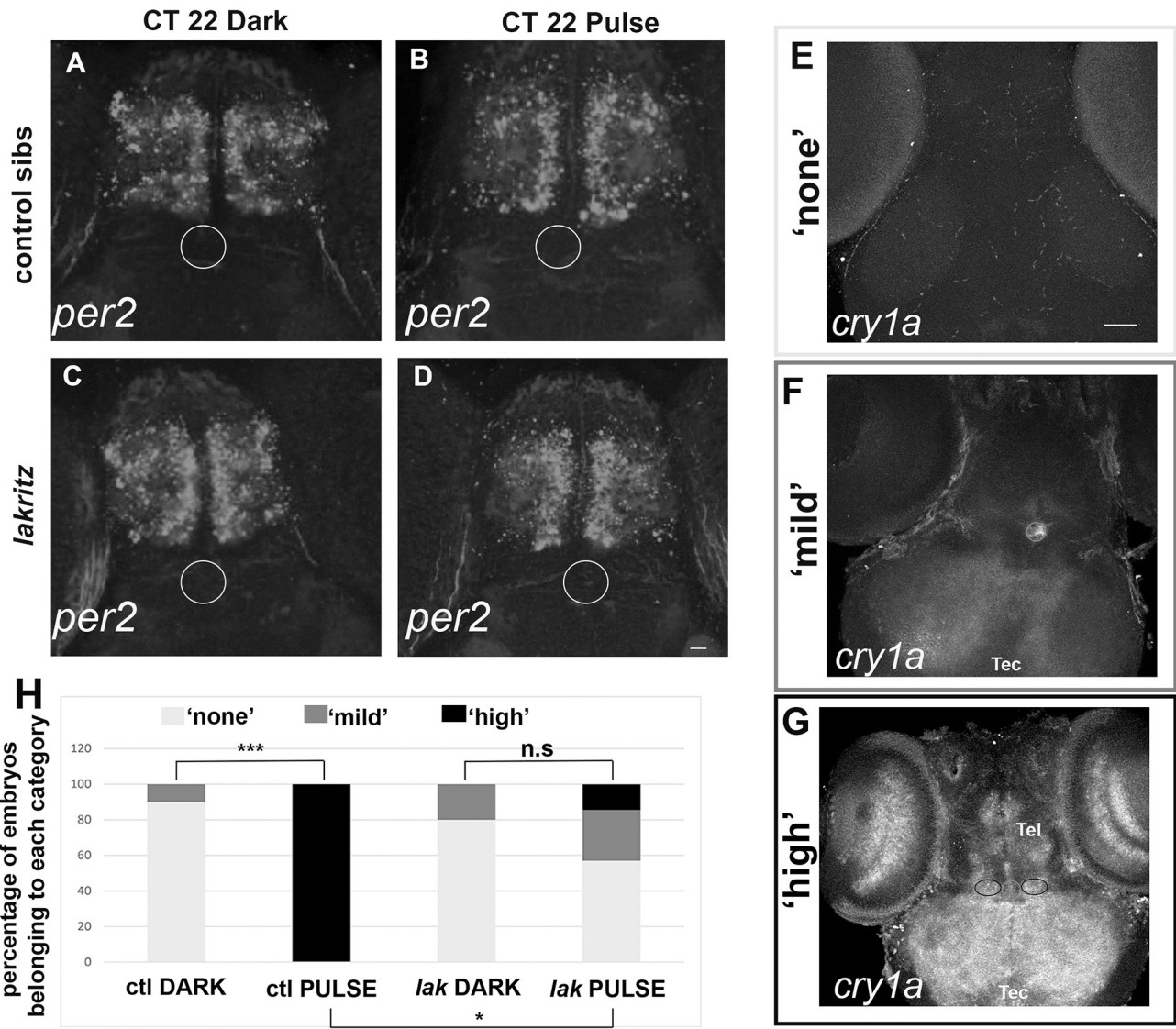

**Fig 3.** *cry1a* but not *per2* expression is induced upon a pulse of white light at CT21 (A-G) Expression of *per2* (A-D*)* or *cry1a (*E-G*)* at CT22 in 7 days old larvae. Before fixation for *in situ* at CT 22 the larvae were treated exactly like in the experiment described in Fig 2D, meaning that they were not depigmented before the light pulse which was administered from CT21 and CT22 (CT 22 pulse). In parallel, larvae from the same litter were maintained in the dark and fixed at CT22 (CT 22 dark). (A-D) The white ellipse identifies the pineal gland where no expression is observed. ctl DARK: n = 3, ctl PULSE n = 7, *lak* DARK: n = 4, *lak* PULSE: n = 7. Scale bar: 50 μm. (E-G) Three different expression patterns can be identified with the *cry1a* probe, The grey ellipse surrounds the pineal which expresses cry1a in the 'mild' and 'high' patterns, the two dark ellipses in G surrounds the habenulae which express *cry1a* in the 'high' pattern. Tel = Telencephalon, Tec = Optic tectum. Scale bar: 200 μm. (H) Countings of the repartition of Dark and Pulsed 7 days old larvae at CT22 stained with the *cry1a* probe, ctl DARK: n = 10, ctl PULSE n = 4, *lak* DARK: n = 5, *lak* PULSE: n = 7. Ctl DARK versus Ctl PULSE: p = 0.0010, *lak* DARK versus *lak* PULSE: p>0.9999, ctl DARK versus *lak* DARK: p>0.9999, ctl PULSE versus *lak* PULSE: p = 0.0182 using Fisher test.

### *opn4xa* function contributes to endogenous period setting in LL

As circadian rhythms of locomotor activity are established and photoentrain in absence of RGCs, albeit with less efficiency, we wondered if *opn4xa+* projection neurons present in the pineal gland could play a role in the establishment and photoentrainment of circadian rhythms [35]. We, thus, generated a mutant allele for *opn4xa* via CRISPR/Cas9 genome editing using guide RNAs targeting the second coding exon. Amongst various alleles that were generated,

we selected an allele that displays a 17 nucleotides insertion for further analysis (Fig 4A). The protein encoded from this allele is predicted to contain a premature stop codon (S2A Fig) leading to a truncation of the protein in the middle of the second transmembrane domain (S2B Fig), which should result in a protein devoid of a G protein interaction domain. We have verified that the two alternative ORFs also generate a truncation from the mutated allele (S2C Fig) and that the use of an alternative ATG present in both the wt and the mutant allele also leads to a truncation (S2D Fig). Finally, we have used Spliceator as to verify that the mutation does not generate alternative splicing sites (Spliceator (lbgi.fr)) We therefore predicted production of a null allele. Homozygous animals were viable and fertile.

We first looked at the expression of *opn4xa* in the retina of *opn4xa*+/+ and *opn4xa* -/- larvae. The RGC layer contained an average of 33 to 44 *opn4xa*+ cells (S3A–S3E Fig), which number did not significantly vary between the different time points suggesting an absence of diel rhythm in this layer. Surprisingly, we observed a previously undescribed expression of *opn4xa* in the interneuron layer with numerous cells at 4dZT0 and 5dZT0 and very few cells at the other ZT time points (S3A–S3D and S3F). The number of *opn4xa*+ cells in both the RGC and the interneuron layer are normal in the *opn4xa* -/- retina (Figs Fig 4B, 4C, S4A and S4B RGC layer: *opn4xa* +/+: 26.17 ± 7.33, n = 6; *opn4xa*-/-: 23.8±6.92, n = 10; p = 0.42 using a Mann-Whitney two-tailed test; interneurons layer: *opn4xa* +/+: 26.33± 11.5; *opn4xa*-/-: 22.9±7.19; p = 0.63 using a Mann-Whitney two-tailed test). Similarly, we have observed that the number of *opn4xa* + pineal cells were similar in *opn4xa* +/+ and *opn4xa* -/- larvae (Fig 4D and 4E (*opn4xa* +/+: 4.33±1.51 n = 6, *opn4xa*-/-: 4.75± 1.98 n = 9, p = 0.85 using a Mann-Whitney two-tailed test). In addition, pineal *opn4xa*+ cells express the Wnt effector *tcf7* [35]. At 6 days post fertilization, the *opn4xa*-/- pineal gland displayed normal expression of *tcf7* (*opn4xa* +/+: 5.5 ± 2,5 (n = 4), *opn4xa*-/-: 6± 3 cells (n = 3); mean ± S.D; S4C and S4D Fig). Upon illumination with a 30 min pulse of light, *fos* is expressed in 2–5 cells of the pineal gland (Fig 4D). We have previously shown that this expression corresponds to *opn4xa*+ cells [35]. On the other hand, *fos* expression is virtually absent in the pineal gland of *opn4xa*-/- embryos after a 30 min pulse of light (Fig 4D; at 3 days *opn4xa* +/+: 4.8 ± 0.8 (n = 9), *opn4xa*-/-: 0.25 ± 0,7 cells (n = 8) per pineal; at 7 days *opn4xa* +/+: 5.9 ± 3,5 (n = 9) *opn4xa*-/-: 0.4 ± 0,7 cells (n = 13); mean ± S. D). Altogether our results suggest that *opn4xa*-/- larvae contained normal numbers of *opn4xa* + cells in the retina and the pineal gland but have lost expression of *fos* following a 30 mn light pulse. We therefore used the *opn4xa* mutant as a model in which the pineal ON response is impaired.

To identify a role for *opn4xa* in the control of circadian rhythms, we analysed locomotor behaviour of *opn4xa*-/- larvae under various illumination regimes. Similar to the analysis we performed in *lak* larvae, for each of these conditions, three independent experiments were performed. Within each independent experiment, the same number of *opn4xa* -/- and *opn4xa* +/+ siblings were randomly selected and the mean of the three experiments was plotted. We found that *opn4xa* -/- larvae still entrained to LD cycles and did not show any difference in levels of locomotor activity when compared to their *opn4xa* +/+ siblings (Fig 5B and S5 Table). In addition, *opn4xa*-/- larvae were able to maintain rhythms of locomotor activity with a similar frequency and period as wild-type larvae in DD (Fig 5C and 5D and S6 Table). *opn4xa*-/- larvae placed in LL still showed circadian rhythms of locomotor activity (Fig 5E and 5F) but with several alterations. First, *opn4xa*-/- larvae were significantly more active during the first night (S8 Table). More importantly, in LL the period was shorter for *opn4xa*-/- larvae compared to controls using both the FFT-NLLS (*opn4xa*+/+: 25.31 ± 3.29 hours (n = 65), *opn4xa*-/-: 24.71 ± 3.32 hours (n = 64); p = 0.041; Mann-Whitney two-tailed test) and mFourfit methods (*opn4xa*+/+: 25.84 ± 1.60 hours (n = 66), *opn4xa*-/-: 25.12 ± 1.739 hours (n = 66); p = 0.012). Altogether these results suggest that *opn4xa* contributes to endogenous period setting in LL.

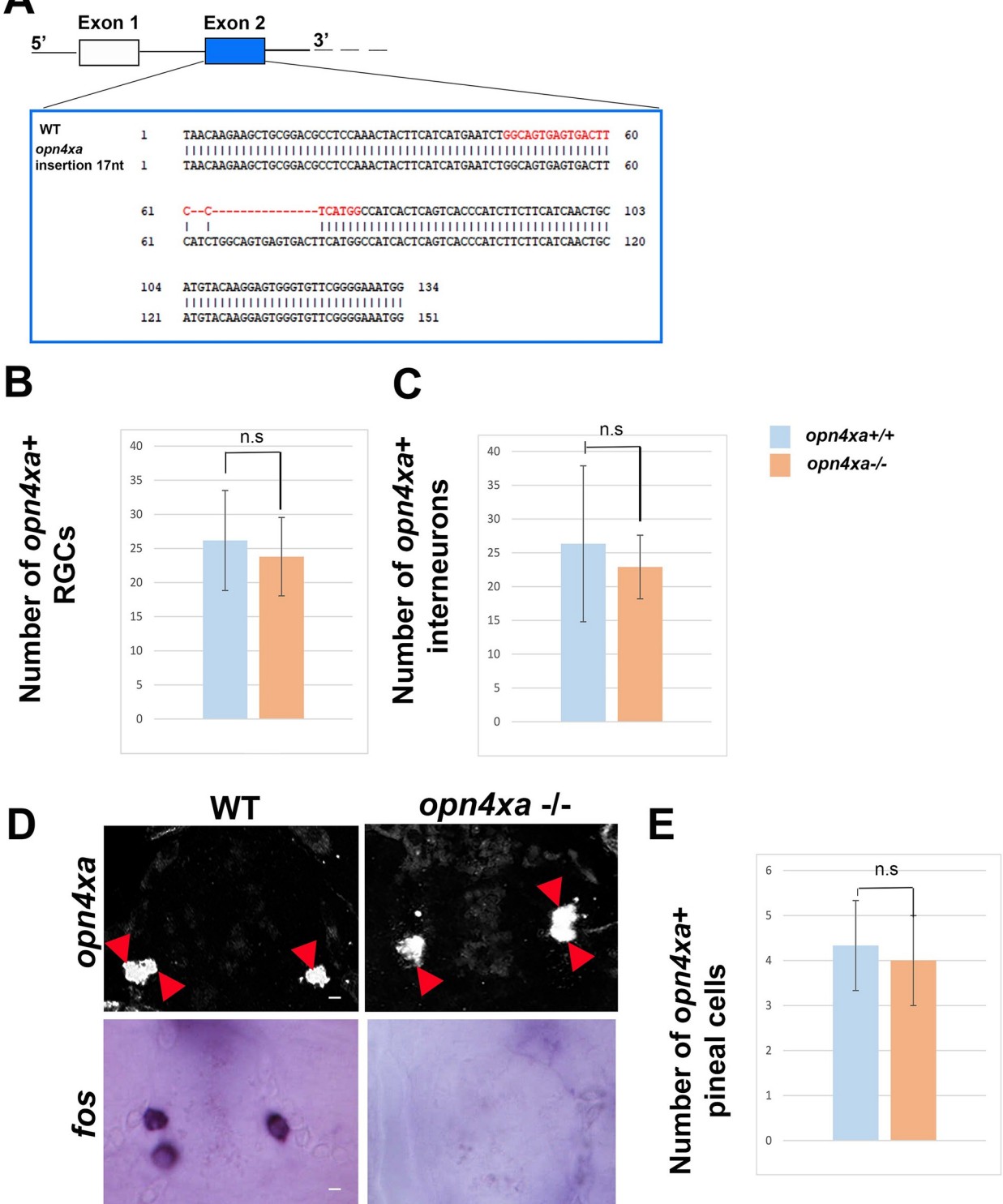

**Fig 4. Mutation in *opn4xa* abolishes light sensitivity in pineal *opn4xa*+ cells** (A) Scheme showing the 5' part of the *opn4xa* locus and in particular the second exon targeted by the CRISPR guide RNA (target sequence is highlighted in red) as well as the WT and mutant exon2 sequences. (B-C) Counts of the number of *opn4xa*+ in the RGC layer (B) and the interneuron layer of the retina (C) after in situ hybridization at 4 days (ZT0) (D) Expression of *opn4xa* in the pineal at 4 days (ZT0) (upper panel) and expression of *fos* at 3 days after 30 min of illumination (lower panel) Red arrowheads point to individual labelled pineal cells. Dorsal views are shown, anterior is up (E) Counts of the number of *opn4xa* + pineal cells. Scale bars respectively represent 10 μm (B) and 5 μm (C-D). n.s: not significant, see results section for details.

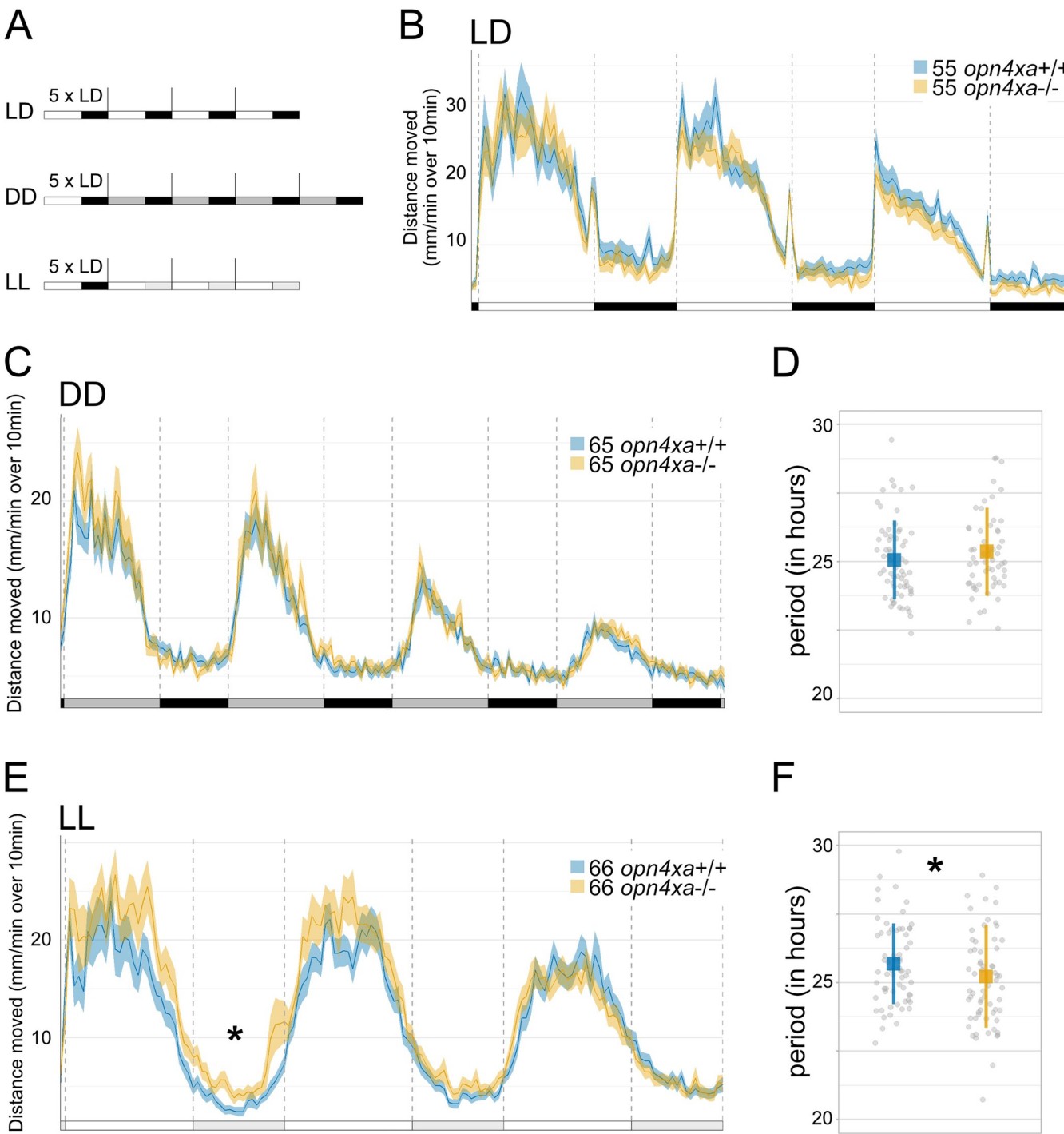

**Fig 5. Locomotor activity of larvae devoid of *opn4xa*-mediated photosensitivity (*opn4xa*-/-) in LD, DD and LL. A)** Experimental design of LD, DD and LL experiments. White rectangles represent the day period, black rectangles represent the night period, dark grey rectangles represent the subjective day period and light grey rectangles the subjective night. For each experiment, larvae are entrained for 5 LD cycles and their locomotor activity is tracked either in LD (LD), constant darkness (DD) or constant light (LL) the larvae are therefore 5dpf at the beginning of locomotor activity measurements. **B)** Average distance moved (mm/min over 10min) of 3 independent experiments in LD. Mean ± SE. The distance moved is not different in *opn4xa*+/+ and *opn4xa*-/- larvae during the 1st (p = 0.73), 2nd days (p = 0.50) and 3rd days (p = 0.07) nor during the 1st (p = 0.30), and 2nd nights (p = 0.27) (S4 Table). A lower level of activity is found in *opn4xa*-/- larvae during the 3rd night (p = 0.01) but is visually clear in only one of the 3 independent experiments (Mann-Whitney two-tailed test). **C)** Average distance moved (mm/min over 10min) of 3 independent experiments in DD. Mean ± SE. **D)** Estimation of the periods using the FFT-NLLS method calculated over four cycles. The mean period is not significantly different between control and *opn4xa*+/+ and *opn4xa*-/- larvae in DD (*opn4xa*+/+: 25.05 ± 1.43 hours (n = 64), *opn4xa*-/-: 25.35 ± 1.60 hours (n = 60); mean±SD; p = 0.29; Mann-Whitney two-tailed test). Mean± sd (in hours) is represented. Each grey point

represents a larva. **E)** Average distance moved (mm/min over 10min) of 4 independent experiments in LL. Mean ± SE. *opn4xa-/-* are more active than controls during the first night (p = 0.02, see S6 Table). **F)** Estimation of the periods using the FFT-NLLS method calculated over three cycles. The mean period is significantly different between *opn4xa+/+* and *opn4xa-/-* larvae in LL. Mean± sd (in hours) is represented. Each grey point represents a larva.

Since *opn4xa-/-* larvae showed a slight hyperactivity during the first night as well as a subtle period phenotype in LL, we began to analyse the possible molecular mechanisms behind these effects using RTqPCR (Fig 6). Analysis of clock gene expression in *opn4xa-/-* versus wt background reveal normal rhythms of clock genes expression in LD except for a slight increase in *tefa* and *cry1a* in *opn4xa -/-* larvae (Fig 6B and 6D). In LL after training, the expression of *bmal1a* was higher in *opn4xa -/-* than in wt larvae (Fig 6A). Interestingly, the expression of some Bmal1a direct targets (*cry1a*, *dec1*) was also increased (Fig 6B and 6C) while expression of *nr1d2a*, *per1a*, *per1b*, *per2* remain unchanged (Fig 6E-H). Overall, only subtle and specific molecular phenotypes were detected in *opn4xa -/-* larvae in LL. In addition, their relation to the locomotor activity phenotypes is not clear.

### *opn4xa* function is dispensable for photoentrainment to a pulse of white light during the subjective night

We next assessed the ability of pulses of white light at CT 16 and CT21 to induce phase shifts in an *opn4xa-/-* background. We observed that under such conditions *opn4xa-/-* larvae shift their activity to the same extent as their wild-type siblings (Fig 6 and Table 2). Furthermore, *opn4xa-/-* larvae did not show any difference in the level of activity during the pulses of light at CT16 or CT21 compared to wildtype siblings, implying that photosensitivity controlled by *opn4xa* is not required for masking (CT16, Fig 7B, *opn4xa+/+*: 20.30 ± 10.03 mm/min over 10min (n = 58), *opn4xa-/-*: 19.42 ± 10.76 mm/min over 10min (n = 58); p = 0.56; Mann-Whitney two-tailed test. CT21: Fig 7D, *opn4xa+/+*: 12.85 ± 6.80 mm/min over 10min (n = 44), *opn4xa-/-*: 16.54 ± 8.55 mm/min over 10min (n = 44); p = 0.47). These results show that the intrinsic photosensitivity of *opn4xa* expressing cells is not necessary for circadian photoentrainment or masking.

Since neither the absence of RGCs (Fig 2) nor the loss of *opn4xa*-dependent photosensitivity (Fig 7) abolished the capacity of larvae to photoentrain to pulses of light performed in the early or late subjective night, a possible compensation could occur between RGCs and *opn4xa* expressing cells of the pineal gland. To begin addressing this question, we tested photoentrainment properties of *lak-/-; opn4xa-/-* larvae (referred to as 'double'). Compared to *lak* simple mutants, double mutants did not show an attenuated phase shift response to pulses of light at CT16 or CT21 (Fig 8). This suggests that other photosensitive cells mediate photoentrainment in zebrafish.

Finally, *lak* and *lak/opn4xa* double mutant larvae show similar levels of activity during the light pulse both at CT 16 and CT21 (CT16: *lak*: 18.23 ± 8.65 mm/min over 10min (n = 27), double: 19.79 ± 12.39 mm/min over 10min (n = 27); p = 0.94, CT21: *lak*: 17.46 ± 6.65 mm/min over 10min (n = 27), double: 16.8 ± 8.05 mm/min over 10min (n = 27), p = 0.42, Mann-Whitney two-tailed test). This suggests an absence of redundancy between RGC and *opn4xa* photosensitivity for masking of locomotor activity.

## Discussion

Experiments performed in mouse have established a role for melanopsin expressing cells of the eye in mediating light input to the circadian rhythm in mammals. In addition to the eye, non-mammalian vertebrates exhibit several extraocular sites of photoreception, raising the

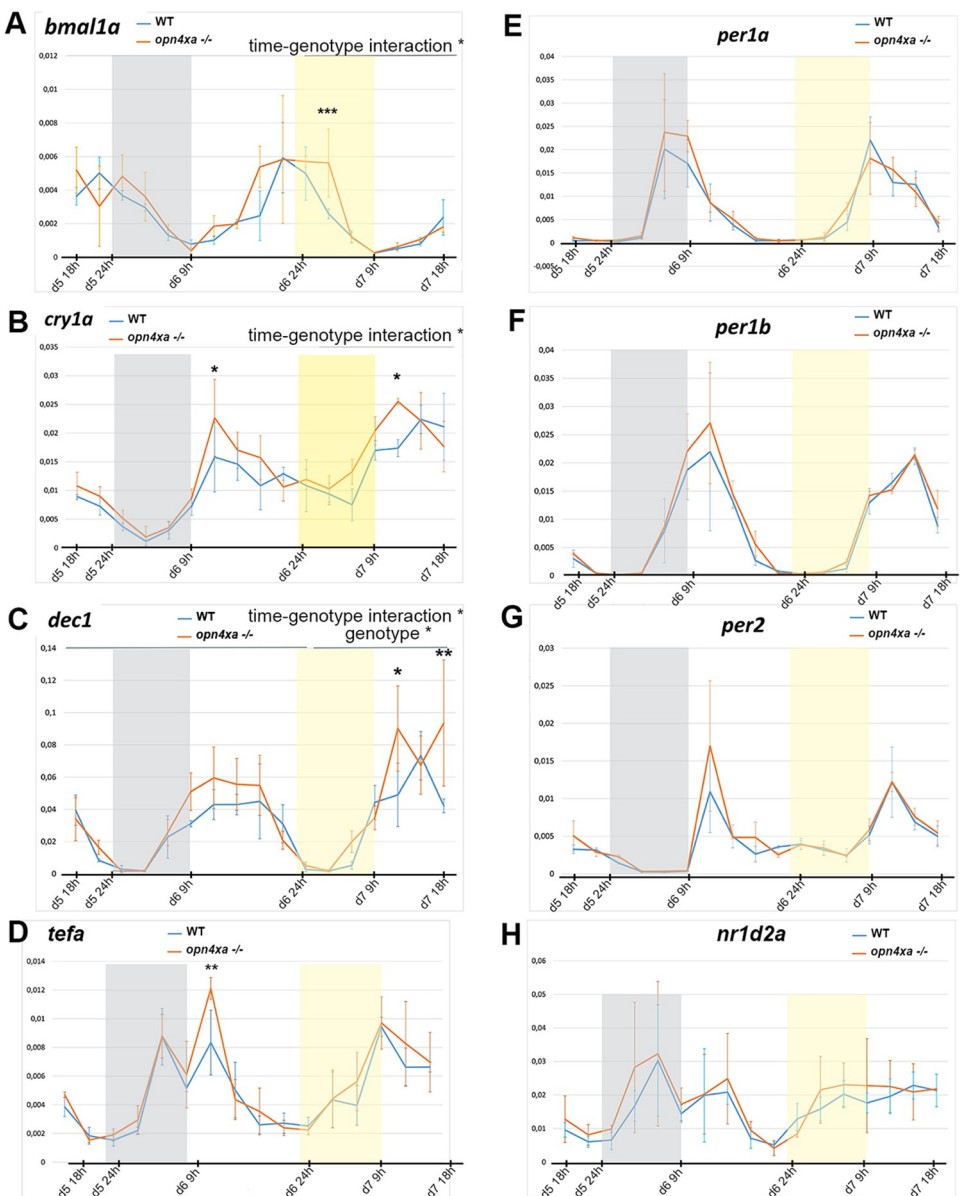

**Fig 6.** *opn4xa* -/- **larvae show subtle modifications of a few clock genes in LL: RTqPCR performed on pools of 15 larvae for the gene indicated at the top of the figures.** Mean expression relative to beta actin ± s.d. Three pools of larvae were used for each time point. 'wt' refers to pool of larvae from crosses of *opn4xa*+/+ animals (siblings of the *opn4xa*-/- fishes used for the *opn4xa*-/-points). Larvae were exposed to LD cycles (until d6 21h) followed by a LL cycle (from d6 21h to d7 18h). The grey rectangles represent the night phase, the yellow rectangles represent the subjective night in the LL cycle. The data were analysed using two-way ANOVAs which revealed time-genotype interaction for *bmal1a* and *cry1a*, as well as statistical differences between genotypes for specific time points using Bonferroni post-hoc tests. p< 0.05; ** p< 0.001; ***p< 0.0005.

question of the relative impact of those different inputs. Herein we show that circadian rhythms of locomotor activity are established and photoentrain in the absence of RGCs in zebrafish larvae. Furthermore, our results show that the absence of a functional eye affects masking, but in a circadian dependent manner. As zebrafish also possesses melanopsin expressing cells in their pineal gland [35], we engineered an *opn4xa* mutant line to address the role of *opn4xa*-dependent photosensitivity in this structure. Our data suggests that *opn4xa* is neither

**Table 2. Quantification of the phase shifts in *opn4xa*+/+ versus *opn4xa*-/- larvae kept in DD or submitted to a 2 hours pulse of white light at CT16.** As for Table 1, the Δphase is the difference between the phase of the two last cycles and the phase of the two first cycles. A Phase shift was observed in DD owing to the period that is close to 25 hours which generates a ~1 hour-shift every cycle. Phases were calculated with the FFT-NLLS method. Upon a pulse of light at CT16 or CT21 a statistical difference was observed between DD and pulsed ctrl larvae (****, p<0.0001 using a Mann-Whitney two-tailed test). In both phase delays (PD, pulse of light at CT16) and phase advance paradigms (PA, pulse of light at CT21), no statistical difference between *opn4xa*+/+ and *opn4xa*-/- was observed using a Mann-Whitney two-tailed test. For each type of paradigm (DD, PD and PA) three independent experiments were pooled.

| Condition | Δphase Mean±S.D (n) | P value Mann-Whitney two-tailed test |
|---|---|---|
| DD *opn4xa* +/+ | 1.64±2.92 (39) | |
| PD-pulsed *opn4xa* +/+ | 4.73±2.63 (36) | *opn4xa* +/+: PD vs DD: **** |
| PD-pulsed *opn4xa* -/- | 5.23±4.81 (35) | PD *opn4xa* -/- vs *opn4xa* +/+: 0.32 |
| PA-pulsed *opn4xa* +/+ | -2.61±4.05 (60) | *opn4xa* +/+: PA vs DD: **** |
| PA-pulsed *opn4xa*-/- | -2±2.88 (20) | PA *opn4xa* -/- vs *opn4xa* +/+: 0.39 |

involved in masking nor in the establishment/photoentrainment of circadian rhythms. While our analysis does not support a redundant role for RGCs and *opn4xa* photosensitivity during photoentrainment of circadian rhythms it reveals a subtle function of *opn4xa*-dependent photosensitivity, possibly in the pineal, in the control of period length of circadian rhythms in constant light.

## Subtle defects in *opn4xa*-/- mutants in LL

While no differences in period or amplitude of locomotor rhythms are observed for *opn4xa* mutant larvae in DD, subtle alterations are observed in LL. Abrogation of *opn4xa* activity

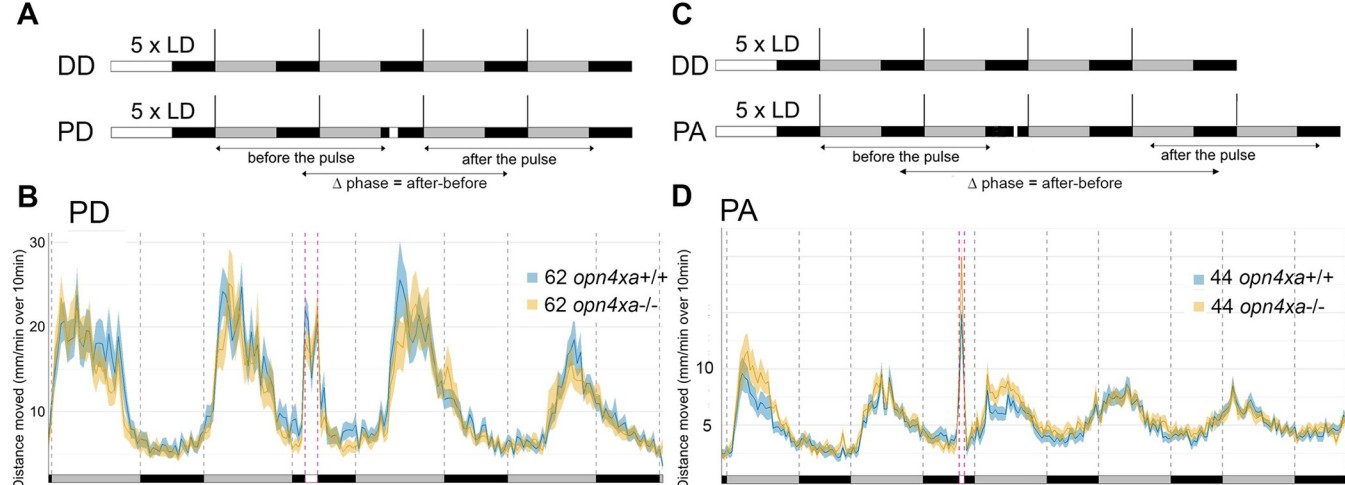

**Fig 7. Larvae devoid of *opn4xa*-mediated photosensitivity (*opn4xa*-/-) still photoentrain to pulses of light at CT16 and CT21. A)** Experimental design of phase shift experiments. White rectangles represent the day or light pulse period, black rectangles represent the night period and dark grey rectangles represent the subjective day. For each experiment, larvae are entrained for 5 LD cycles the larvae are therefore 5dpf at the beginning of locomotor activity measurements. Locomotor activity is tracked either in constant darkness for 4 days (DD) or tracked in constant darkness for 4 days and subjected to a 2-hours pulse of light during the night of the 2nd day of constant darkness (PD). The phase of locomotor activity is calculated for each larva before and after the timing of the pulse for DD and PS experiments and the Δphase (phase after the pulse–phase before the pulse) is calculated. **B)** Average distance moved (mm/min over 10min) of 3 independent PD experiments. Mean ± SE. The Δphase of *opn4xa*+/+ and *opn4xa*-/- larvae calculated with the FFT-NLLS method is not significantly different. *opn4xa*+/+ and *opn4xa*-/-show similar levels of activity during the light pulse (p = 0.56, Mann-Whitney two-tailed test). **C)** Experimental design of phase advance (PA) experiments. The iconography is similar to A). PA-pulsed larvae were subjected to a one-hour pulse of light at CT21. **D)** Average distance moved (mm/min over 10min) of 3 independent PA experiments. Mean ± SE. The Δphase of *opn4xa*+/+ and *opn4xa*-/- larvae calculated with the FFT-NLLS method is not significantly different (Mann-Whitney two-tailed test).

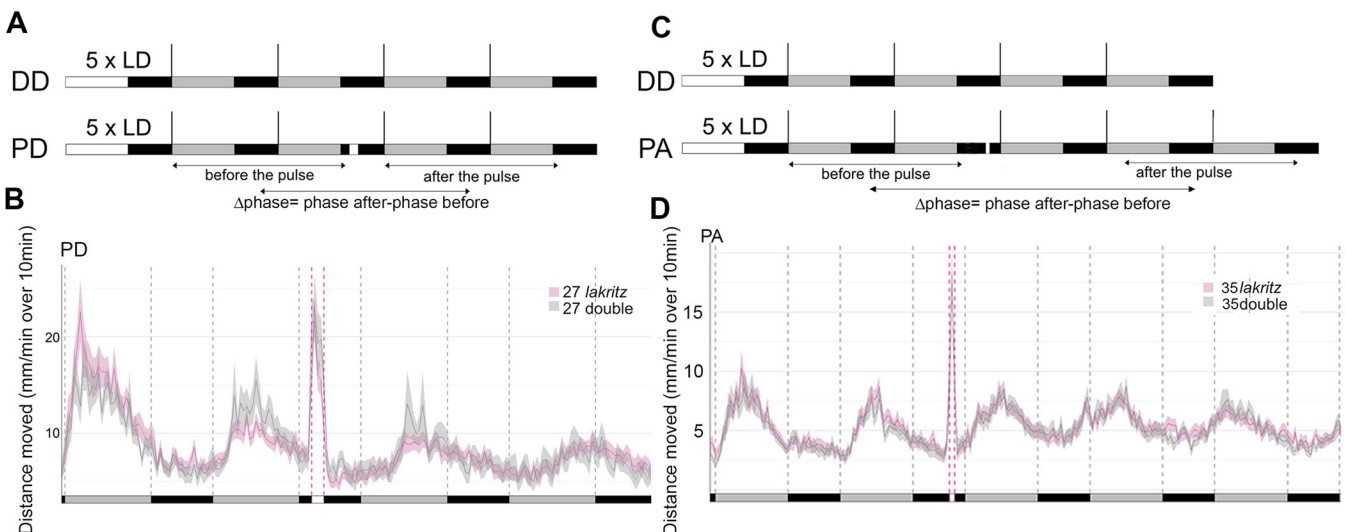

**Fig 8. Larvae devoid of RGCs and *opn4xa*-mediated photosensitivity still entrain to pulses of light at CT16 and CT21. A)** Experimental design of phase shift experiments. White rectangles represent the day or light pulse period, black rectangles represent the night period and dark grey rectangles represent the subjective day. For each experiment, larvae are entrained for 5 LD cycles the larvae are therefore 5dpf at the beginning of locomotor activity measurements. Locomotor activity is tracked either in constant darkness for 4 days (DD) or tracked in constant darkness for 4 days and subjected to a 2-hours pulse of light during the night of the 2nd day of constant darkness (PD). The phase of locomotor activity is calculated for each larva before and after the timing of the pulse for DD and PS experiments and the Δphase (phase after the pulse–phase before the pulse) is calculated. **B)** Average distance moved (mm/min over 10min) of 3 independent PD experiments (n = 27 for *lak* -/- referred as *lak* and n = 27 *lak*-/-; *opn4xa*-/- larvae referred as 'double'). Mean ± SE. The Δphase of *lak* and double larvae calculated with the FFT-NLLS method are not significantly different (see Table 3). **C)** Experimental design of phase advance (PA) experiments. The iconography is similar to A). After 5 training cycles in LD, PA-pulsed larvae were subjected to a one-hour pulse of light at CT21. **D)** Average distance moved (mm/min over 10min) of 3 independent PA experiments (n = 27 for *lak* and n = 27 double larvae). Error bars represent SE. The Δphase of *lak* and double larvae calculated with the FFT-NLLS method are not significantly different (see Table 3).

reduces the increase of period length observed when larvae are placed in LL. A similar defect is observed in *Opn4*-/- mice placed in constant light condition [8,9] suggesting this could be a conserved function of melanopsin. Interestingly, this phenotype is not observed in *lak* mutant suggesting that in zebrafish this melanopsin function might involve the pineal gland rather than the eye. While the effect of *opn4xa* mutation on period length in LL is subtle, such an effect is not at all observed in DD. We see two possible hypotheses. First, *opn4xa* homozygous larvae could sense less light than controls and since period shows a reverse correlation with the intensity of light exposure [44,45], it could result in a lower period. A second hypothesis is that

**Table 3. Quantification of the phase shifts in *lak* versus double mutant larvae kept in DD or submitted to pulses of white light at CT16 or CT21.** As for Tables 1 and 2, the Δphase is the difference between the phase of the two last cycles and the phase of the two first cycles. A Phase shift is observed in DD owing to the period that is close to 25 hours which generates a ~1 hour-shift every cycle. Phases were calculated with the FFT-NLLS method (biodare2.ed.ac.uk). Upon a pulse of light at CT16 or CT21 a statistical difference is observed between DD and pulsed *lak* larvae (****. p<0.0001 using a Mann-Whitney two-tailed test). In both phase delays (PD, pulse of light at CT16) and phase advance paradigms (PA. pulse of light at CT21), no statistical difference between *lak* and double larvae is observed using a Mann-Whitney two-tailed test. For each type of paradigm DD, PD and PA three independent experiments were pooled.

| Condition | Δphase Mean±S.D (n) | P value Mann-Whitney two-tailed test |
|---|---|---|
| DD *lak* | 2.143±2.09 (31) | |
| PD-pulsed *lak* | 5.72±2.32 (37) | *lak*: PD vs DD: **** |
| PD-pulsed double | 4.90±2.23 (14) | PD double vs *lak*: 0.33 |
| PA-pulsed *lak* | -0.57±4.49(29) | *lak*: PA vs DD:**** |
| PA-pulsed double | -3.16±4.9 (16) | PA double vs *lak*: 0.24 |

the *opn4xa -/-* clock is less stable in LL compared to DD. Indeed, we observed a stronger dampening of the circadian rhythms in LL compared to DD especially when four cycles were filmed and despite a moderate light intensity (20 lux). Along the same line, the variation observed for the periods are higher in LL than in DD for both control and mutant larvae suggesting again less robustness in clock activity in constant light conditions.

While we have analysed the expression of clock genes our results did not yield a clear molecular explanation for the *opn4xa -/-* period phenotype in LL. Indeed *opn4xa -/-* larvae only show a modest increase in *Bmal1a*, *cry1* and *dec1* expression in LL (Fig 5). A loss of *dec1* using morpholino antisense technology suggests a role for this gene in controlling circadian rhythms of locomotor activity induced by a light pulse but a role for this gene in period control in LL has not been addressed [46]. Along the same line a double mutant line for *per2* and *cry1a* exhibit deficits in the generation of rhythms following a light pulse but the role of *cry1a* during period control in LL has not been addressed [47]. In contrast, *per2* regulates period length in constant light but its transcription is not affected in *opn4xa-/-* larvae ([48,49]; Fig 6G)); it is however possible that affecting the level of CRY1A could modify PER2 activity therefore leading to the observed phenotype. Further experiments will be required to address this question.

## RGCs, but not *opn4xa*, are involved in masking

Compared to their control siblings, we found *lak* mutant larvae to be less active during the light phases of LD cycles as well as when subjected to a pulse of light at CT16 but not at CT21. This reveals a role for RGCs in positive masking in zebrafish larvae as well as a circadian control of this masking activity. It is important to note that while we refer to this as an effect of "masking", it is at this step impossible to distinguish whether the *lak* mutation impairs the arousing effect of visual stimuli or the arousing effect of light itself. Moreover, masking is not completely abolished in *lak* mutant larvae: in LD as well as during a light pulse at CT 16, *lak* mutant larvae still show some masking, albeit at a reduced level and masking is not affected in these larvae at CT21. Could *opn4xa* photosensitivity from the pineal compensate for the lack of RGCs? *opn4xa -/-* larvae display no defect in masking of locomotor activity in LD or during a pulse of light suggesting that *opn4xa*-dependent photosensitivity is dispensable for this type of masking. In addition, *lak*; *opn4xa* double mutants show a similar activity to *lak-/-* larvae during a pulse of light at CT16 and CT21, suggesting that there is no redundancy between the eye and *opn4xa+* cells in the pineal for masking control. Other photosensitive cells are thus involved in this process. Among these could be the classical photoreceptors of the pineal or deep brain photoreceptors, such as those involved in the locomotor response to a loss of illumination [50].

## A Differential Role for RGCs in controlling phase advance versus phase delays in the rhythms of locomotor activity?

*lak* mutant larvae show a reduced phase-advance but no defect in a phase delay paradigm which could indicate a specific role for the RGC in controlling photoentrainment in a phase-advance context. On the other hand, *lak* mutant larvae display less aggregated pigment granules than siblings (S5 Fig). Could this impact the way light penetrates their body? While this remains a possibility, we do not think it is a very likely hypothesis since 1. even in *lak* mutant there are rather wide pigment-free areas over the fish brain and in particular over the pineal (S5B Fig) 2.light can still penetrate from the side of the body as pigments are found only on the dorsal and ventral most epidermis (S5C Fig). Experiments aiming at targeting different RGC subpopulations will help consolidate the role of the eye in controlling phase advances as well as identifying downstream circuits.

Finally, our results show an induction of *cry1a* upon a pulse of light at CT21 and a strong reduction of this induction in *lak* larvae, which suggests an involvement of this gene in the molecular mechanism of phase advance induction, as previously suggested for the molecular rhythms in cell cultures [39]. In contrast, *per2* was not induced in our phase advance paradigm despite previous evidences for *per 2* induction by light at different developmental stages and circadian time [43]. Our results thus highlight an additional layer of complexity in the mechanisms of *per2* regulation by light.

## RGCs and *opn4xa* are largely dispensable for shifting circadian rhythms of locomotor activity in response to a pulse of white light

The present study shows that neither the eye nor *opn4xa* mediated photosensitivity in the pineal gland is absolutely required for the development of circadian rhythms and circadian photoentrainment. The absence of a strong requirement for the eye to control the circadian system in zebrafish is surprising given that *Astyanax mexicanus* blind cavefishes are arrhythmic in DD [27] while *Phreatichthys andruzzii* adult cavefishes, which also exhibit a complete eye degeneration, and are arrhythmic in LD when fed at random times [28]. In light of our data, we propose that apart from the eye, other photosensitive structures might be affected in these other fish species. This in turn brings the question of which structure(s) relay light information to control circadian rhythm in fishes and other non-mammalian animals? The pineal gland, with its classical photoreceptors and *opn4xa+* projection neurons is an appealing candidate [51]. Strategies aiming at genetically killing this structure or impairing its activity will surely help unravelling its function. Studies describing the effect of surgical pinealectomy have been reported in a number of non-mammalian vertebrates. The phenotypes induced seem to depend strongly on the species. For instance, pinealectomy abolishes rhythms in the stinging catfish but not in the amur catfish or the lake chub. Interestingly, in species where rhythms are maintained upon pinealectomy a change in period can occur (see [52] for a review). A similar variety of phenotypes are induced upon pinealectomy in reptiles or birds. In addition to the pineal gland, reptiles have a parietal eye, a structure that is developmentally and spatially related to the pineal gland. Interestingly simultaneous removal of the eye, the pineal gland and the parietal eye in two species of lizards (*P. Sicula* and *S. olivaceous*), does not impair rhythms of locomotor activity while on the contrary these rhythms are lost if in addition to this triple ablation injection of dark ink between the skin and the skull is performed [53]. Similarly, experiments in songbirds suggest the existence of additional photosensitive structures located in the brain that control photoentrainment [54]. Altogether these results highlight the existence of other brain structures mediating light inputs on the circadian system. Interestingly, melanopsin expression has been described in other brain areas in the zebrafish larva: *opn4a* is expressed within the presumptive optic area, *opn4b* is found in the ventral forebrain and the thalamic region, and *opn4.1* is detected in a specific domain located in the ventricular region at the junction between the caudal hindbrain and the anterior spinal cord [34,50]. Larvae double mutant for *opn4.1* and *opn4xb* show a decreased locomotor activity during the day but no circadian phenotype [55]. As 42 opsin genes are predicted in the zebrafish genome, of which 20 are expressed in the adult brain [56], further characterization of their expression in the larval brain will be needed to define the best candidates for further study. Finally, the possibility remains that photoentrainment in zebrafish occurs as a result of direct photosensitivity of motoneurons or muscles themselves as all cells and organs have been shown to be directly photosensitive and light-entrainable in this species (see [57] for a review).

Taken together, our results highlight profound differences in the establishment and photoentrainment of the circadian system between the diurnal zebrafish and other species such as

mice and human. A crucial, yet open question is whether these divergences reflect the different phylogeny of these species or their different use of temporal niches. The photosensitive capabilities of the zebrafish in particular and of aquatic species in general (as judged by the number of opsins predicted in the genome) far exceed that observed in mammals. This could imply a greater level of complexity and robustness in circadian control in zebrafish independently of its temporal niche. However, the human brain also expresses opsins (*OPN3* and *OPN5*; [58,59]) suggesting the existence of deep brain photoreceptors in diurnal primates and the possibility that they participate in photoentrainment.

## Material and methods

### Ethics statement

All animals were handled in the CBI fish facility, which is certified by the French Ministry of Agriculture (approval number A3155510). The project was approved by the French Ministry of Teaching and Research (agreement number APAFIS#3653–2.016.011.512.005.922).

### Zebrafish lines and developmental conditions

Embryos were reared at 28 degrees in a 14h light/10h dark cycle with lights on at 9:00 and lights off at 23:00.

The *lak* mutant line has been described previously [32], *lak* homozygous mutants were identified by their dark coloration. Genotyping of *lak* individuals was performed as previously described [32].

### Generation of an *opn4xa* mutant allele

An *opn4xa* mutant allele was generated using the CRISPR/ Cas9 targeted genome editing. For this. a target site was designed in the second exon by manual screening for PAM sites. Transcription of the guide and coinjection of the guide mRNA with cas9 mRNA was performed as described in [60]. Screening of potential mutants was performed using T7 endonuclease (NEB) treatment of PCR products amplifying the second *opn4xa* exon (Fw: 5' CACAACAT AAACTGTAACTGCATCC 3', Rev: 5' GACACGGGTATGACACTCAGGAAGG 3'). PCR products from potential carriers were subsequently subcloned and sequenced. In this manner we identified several interesting carriers among which an individual transmitting an allele bearing 17 extra nucleotides in the second exon leading to a premature interruption of the coding sequence.

To genotype *opn4xa* individuals, we used a classical PCR protocol with the following oligos: 5'-GGACGCCTCCAAACTTC-3' (Forward) and 5'-CGAACACCCACTCCTTGTAC-3' Reverse). PCR products of different sizes were obtained (110bp for the wt allele and 127bp for the mutant allele) and resolved on a 4% agarose gel.

### Locomotor activity assays

Larvae zebrafish coming from heterozygous incrosses were raised on a 14:10 hr light:dark cycle at 28°C in Petri dishes with no more than 50 larvae per Petri dish in a water bath inside the fish facility. On the morning of their 5th day of development (9:15–10:30), individual larvae were placed in each well of a 96-well plate containing aquarium fish water and placed back in the water bath. Light intensity during the entrainment was 110 lux at 6500K. On the evening (16:00–20:00), the plate was put in an hermetic box in a dark room maintained at 27°C with a heater. This home-made box [61] was continuously illuminated from below with two panels of infra-red lights as well as neutral white light (4000K, 20 lux at water surface)) controlled by a

timer from 9:00 to 23:00. Larvae were then filmed at 30 frames per second, with a ceiling mounted infra-red camera connected to a computer on the following days (from the 6[th] day of development to the 9[th] or 10[th] day of development) in controlled conditions of illumination. The temperature inside the box was monitored using an electronic programmable device (I Button. Maxim). After the experiment, larvae were either genotyped by PCR for *opn4xa* and/ or *lak* and/or simply identified for the *lak* mutation using the dark coloration phenotype. In addition, larvae presenting developmental defects were discarded from the subsequent analysis. Experiments in which too many larvae presented development issues or where temperature issues were present were discarded. At least three experiments were made for each type of assay.

## In situ hybridization

In situ hybridization was performed as described previously [62]. *opn4xa*, *tcf7*, *cry1a* and *c-fos* probes have been described (34,35,62,63). A *per2* probe encompassing the entirety of the second exon was engineered using PCR with the following oligos: Forward primer: 5'-AAATCC GAGTGTCCGTCTGC-3', Reverse primer with T7: 5'-TAATACGACTCACTATAGGGT CTTGTTGCTTCCCGATGAC-3' followed by T7 transcription.

## Locomotor activity analysis

After the experiment, the distance travelled per minute was extracted for each larva using the Ethovision XT13.0 (Noldus. Wageningen. the Netherlands) with the following parameters: for Detection Settings: dynamic substraction; subject color compared to background: Darker; Dark: 7 to 210; Frame Weight: 2; for Track Smoothing Profiles: Minimal Distance Moved: 0.2mm—Direct (A>MDM); for Data Profiles: Results per time bin. Ignore last time bin if incomplete; for Analysis Profiles: Distance moved of the center-point. The obtained files were then analysed using the wakefish program (written in python by L.Sanchou) to extract an average activity of mm/min over 10min for each larva ('DM10 files'). For each experiment, the same number of homozygous mutants and wild-type or control larvae were randomly selected. The Biodare software was used to calculate periods and phases for each larva (biodare2.ed.ac. uk). We choose to use the FFT-NLLS to calculate periods and phases on DM10 files after baseline detrending, as advised [63]. The parameters used for period calculation were as follows: baseline detrending, expected periods from 18 to 30 hours. analysis method FFT-NLLS. The parameters used for phase calculation were as follows: baseline detrending. FFT-NLLS, phase by fit, absolute phase to window. Windows used to calculate the phase of locomotor activity "before the pulse" and "after the pulse" encompass time points from CT0 to CT15 (corresponding from 9am of the 1[st] day in of the experiment to midnight between the 2[nd] and 3[rd] day of the experiment for "before the pulse" and from 9am of the 3[rd] day of the experiment to midnight between the 4[th] and 5[th] day of the experiment). Locomotor activity levels were calculated from the DM10 files by calculating means of the average activity in mm/min over 10 min over a given period for each larva. Statistical analysis was done using Prism. Graphs were generated using R studio (ggplot2 and rethomics packages [64,65]).

## Analysis of clock gene expression using RTqPCR

For each specific stage, pools of 15 larvae were collected and extracted with TRIzol. Reverse transcription and qPCR were performed using a standard protocol [66] with the oligos detailed in S7 Table. All the experiments were performed in triplicates and the mean expression relative to beta actin was calculated. Three pools of larvae were used for each time point.

## Supporting information

**S1 Fig. (A)** Average distance moved merged from PA experiments in 10 min bins. Mean ± SE. The original data is the same than in Fig 2 but here, only the larvae for which a phase could be extracted for the two first and the two last cycles were included in the average. In addition, two rounds of smoothing each using ten successive time points were applied as this made the difference in phase shift between the *lak* and the sib larvae easier to visualize.
(TIF)

**S2 Fig. (A)** Prediction of the protein sequences produced by the wt and mutant exon 2. The part corresponding to the second transmembrane domain (34) is underlined. The red asterisk indicates a premature stop codon. **(B)** Models of the WT and mutant predicted OPN4XA proteins. **(C)** Using alternative ORFs leads to a premature stop codon (indicated with a star). **(D)** The use of an alternative ATG (bold) also leads to a truncated protein.
(TIF)

**S3 Fig. *opn4xa* is expressed in zebrafish RGCs and interneurons. (A-D)** Expression of *opn4xa* at 4days at different ZT using fluorescent in situ hybridization. Lateral view of mounted eyes imaged under the confocal microscope. The ventral side is oriented towards the left upper corner. Based on position, the *opn4xa+* cells from the interneuron layer are most likely horizontal and amacrine cells. Scale bar: 10 μm. **E)** Number of *opn4xa+* cells in the RGC layer in 96–128 hpf zebrafish larvae. All data follows a Gaussian distribution. No statistical differences were observed between the different time points using a one-way ANOVA with Bonferroni post hoc test. **F)** Number of *opn4xa+* cells in the interneuron layer in 96–128 hpf zebrafish larvae. The data at 4dZT0 does not follow a Gaussian distribution. * p<0.05. ** p<0.001. *** p<0.0005 using a Kruskal-Wallis test with Dunn's post hoc comparison.
(TIF)

**S4 Fig. Characterization of *opn4xa*-/- retina and pineal glands. (A-B)** Expression of *opn4xa* in the retina of wt and *opn4xa*-/- larvae at 4days ZT0. The ventral side is oriented towards the downward left corner. **(C-D)** Expression of *tcf7* in the pineal gland at 6 days using in situ hybridization in wt and *opn4xa*-/- larvae. Dorsal views are shown. Anterior is up. Scale bar: 10 μm.
(TIF)

**S5 Fig. Live *lak* and sibling larvae at 5 dpf showing the differences in pigmentation. (A-B)** Dorsal views. **(C)** side views. The red arrow in B points to the position of the pineal gland which is not covered by pigments in *lak* and sib larvae. Scale bar: 0.5 mm.
(TIF)

**S1 Table. Activity of *lakritz* -/- versus control larvae in LD** showing the average distance travelled (mm/min) over a 10 min window averaged during the day (D) or the night (N) periods. Mean ± S.D. D1 corresponds to the first day. The p value and statistical significance using a two-tailed Mann-Whitney test is indicated.
(DOCX)

**S2 Table. Percent of rhythmic *lak* and sib larvae in DD and LL.** The difference between the ctl DD and the *lak* DD populations is not significant using a khi-two test p = 0.957, similar results are obtained between the ctl LL and the *lak* LL, p = 0,91.
(DOCX)

**S3 Table. Activity of *lakritz* -/- versus control larvae in DD** showing the average distance travelled (mm/min) over a 10 min window averaged during the day (D) or the night (N)

periods. Mean ± S.D. D1 corresponds to the first day. The p value and statistical significance using a two-tailed Mann-Whitney test is indicated.
(DOCX)

**S4 Table. Activity of *lakritz* -/- versus control larvae in LL** showing the average distance travelled (mm/min) over a 10 min window averaged during the day (D) or the night (N) periods. Mean ± S.D. D1 corresponds to the first day. The p value and statistical significance using a two-tailed Mann-Whitney test is indicated.
(DOCX)

**S5 Table. Activity of *opn4xa* -/- versus control larvae in LD** showing the average distance travelled (mm/min) over a 10 min window averaged during the day (D) or the night (N) periods. Mean ± S.D. D1 corresponds to the first day. The p value and statistical significance using a two-tailed Mann-Whitney test is indicated.
(DOCX)

**S6 Table. Percent of rhythmic *opn4xa* -/- and ctl larvae in DD and LL.** The difference between the WT DD and the *opn4xa*-/- DD populations is not significant using a khi-two test, p = 0.06.
(DOCX)

**S7 Table. Activity of *opn4xa* -/- versus control larvae in DD** showing the average distance travelled (mm/min) over a 10 min window averaged during the day (D) or the night (N) periods. Mean ± S.D. D1 corresponds to the first day. The p value and statistical significance using a two-tailed Mann-Whitney test is indicated.
(DOCX)

**S8 Table. Activity of *opn4xa* -/- versus control larvae in LL** showing the average distance travelled (mm/min) over a 10 min window averaged during the day (D) or the night (N) periods. Mean ± S.D. D1 corresponds to the first day. The p value and statistical significance using a two-tailed Mann-Whitney test is indicated.
(DOCX)

**S9 Table. qRTPCR primer sequences used in the study.** The right column indicates the ZDB gene ID (https://zfin.org).
(DOCX)

**S1 Data. Numerical data related to Fig 1.**
(XLSX)

**S2 Data. Numerical data related to Fig 2.**
(XLSX)

**S3 Data. Numerical data related to Fig 3.**
(XLSX)

**S4 Data. Numerical data related to Fig 4.**
(XLSX)

**S5 Data. Numerical data related to Fig 5.**
(XLSX)

**S6 Data. Numerical data related to Fig 6.**
(XLSX)

**S7 Data. Numerical data related to Fig 7.**
(XLSX)

**S8 Data. Numerical data related to Fig 8.**
(XLSX)

**S9 Data. Numerical data related to S1 Fig.**
(XLSX)

**S10 Data. Numerical data related to S3 Fig.**
(XLSX)

**S11 Data. Numerical data related to S4 Fig.**
(XLSX)

## Acknowledgments

We are indebted to C. Rampon for allowing us to use the Ethovision Software as well as Julie Batut, Ouria Dkhissy Benyahya and Frederique Gaits-Iacovoni for helpful discussions. We thank M. Halpern, T. Dickmeis and K. Soanes for sharing probes. We would like to thank Brice Ronsin, Stéphanie Bosch and the Toulouse RIO Imaging platform; Stéphane Relexans, Aurore Laire and Richard Brimicombe for taking care of the fish as well as Sophie Polès for technical help.

## Author Contributions

**Conceptualization:** Elise Cau.

**Data curation:** Clair Chaigne, Elise Cau.

**Funding acquisition:** Patrick Blader.

**Investigation:** Clair Chaigne, Elise Cau.

**Methodology:** Dora Sapède, Xavier Cousin, Elise Cau.

**Project administration:** Elise Cau.

**Resources:** Dora Sapède, Xavier Cousin, Patrick Blader, Elise Cau.

**Software:** Laurent Sanchou, Elise Cau.

**Supervision:** Elise Cau.

**Visualization:** Clair Chaigne, Elise Cau.

**Writing – original draft:** Clair Chaigne, Elise Cau.

**Writing – review & editing:** Elise Cau.

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
