## [Decision Letter · Decision Letter 0]

23 Feb 2023

Dear Dr Cau,

Thank you very much for submitting your Research Article entitled 'Analysis of photoentrainment in the diurnal zebrafish suggests a profound divergence with nocturnal rodents' to PLOS Genetics.

The manuscript was fully evaluated at the editorial level and by independent peer reviewers. The reviewers appreciated the attention to an interesting question, but raised some substantial conceptual and experimenta concerns about the current manuscript. Based on the reviews, we will not be able to accept this version of the manuscript, but we would be willing to review a much-revised version. We cannot, of course, promise publication at that time.

If you decide to revise the manuscript for further consideration at PLOS Genetics, please aim to resubmit within the next 60 days, unless it will take extra time to address the concerns of the reviewers, in which case we would appreciate an expected resubmission date by email to plosgenetics@plos.org.

We are sorry that we cannot be more positive about your manuscript at this stage. Please do not hesitate to contact us if you have any concerns or questions.

Yours sincerely,

John Ewer

Academic Editor

PLOS Genetics

Gregory Barsh

Editor-in-Chief

PLOS Genetics

Reviewer's Responses to Questions

**Comments to the Authors:**

Reviewer #1: Chaigne, C et al investigate the role of retinal ganglion cells and a specific melanopsin ortholog, opn4xa, in zebrafish larval circadian locomotion behavior and circadian clock entrainment. They observe relatively minor differences for photic circadian entrainment for lak-/- mutants versus wildtype when the stimulus is given during the end of the night, as well as for total movement under constant light in the case of opn4xa mutants. Double mutants reveal no additional effects on phase entrainment. They also observe interesting additional details for opn4xa expression and possible function, i.e. robust diel changes in interneurons of the eyes other than the RGCs. Their data on fos indicates that opn4xa is the sole mediator of light sensation by the pineal gland at least up to 7dpf. Albeit minor effects, the results provide very valuable additional information on the non-visual functions of an opsin in a non-mammalian vertebrate. Non-mammalian vertebrates possess much higher numbers of opsins than mammals, thus individual loss-of-function effects are expected to be likely small, but nevertheless important to provide to the scientific community interested in understanding how light impacts on animals.

Before the data can be published, the authors should address the following concerns/suggestions:

>As a general layout comment- it would help to include line and page numbers, because it makes it easier to refer to specific sentences in the manuscript.

„The eye in general and melanopsin expressing cells in particular are crucial for circadian rhythms in nocturnal mammals, most notably during photoentrainment, by which circadian rhythms adapt to a changing light environment.“

> Remove nocturnal. This is probably a general feature for mammals, irrespective of nocturnality/diurnality.

„raising the intriguing possibility that vertebrates might employ different molecular/cellular circuits for photoentrainment depending on their phylogeny and/or temporal niche.“

> It is already well established that in zebrafish different peripheral cell types are directly photosensory. See work by the labs of David Whitmore, Nick Foulkes etc.. The authors even refer to the relevant literature in their introduction.

So, this is not raising a possibility, but it is a long-standing fact. The question is more to the mechanistic details, how different photoreceptors contribute to it, and why this ability apparently got lost in mammals. This is where the authors actually make their contributions. They might be not so obvious, but they are nevertheless important for the large picture of non-visual photoreceptor functions. So, it would be appropriate to re-word this summary accordingly.

Introduction:

"Once established, these rhythms persist in constant light conditions, "

> constant conditions, not just “light“ or better re-phase to: “conditions that don’t provide entrainment information, e.g. constant light and temperature.”

"Although ipRGCs mediate circadian and direct effects of light on behaviour in nocturnal mammals, it is unclear if these roles are conserved in diurnal and non-mammalian species"

>The statements after these sentences need serious rewording. It is long established especially by the work of David Whitmore, Nicholas Foulkes and their labs that individual cells in fish are themselves light sensitive, independent of the eyes. The authors refer to Cavallari et al.. 2011, but do not embed the findings and conclusions of this paper well into their introduction. Directly after the citation they write: “While the function of the eye regarding circadian rhythms has not been addressed in fishes…” and “The role of this local photodetection regarding circadian behaviours has not been explored… “.

However, Cavallari et al.. 2011 explicitly states (in their summary!): “We reveal that eye loss does not account for this “blind” clock. Specifically, mutations of two widely expressed non-visual opsin photoreceptors (Melanopsin and TMT opsin) are responsible for the blind clock phenotype in the cavefish.”

Cavallari et al.. 2011 provides evidence for this statement by tissue culture experiments. Interestingly, Cavallari et al.. 2011 have not performed the lak- mutant experiments, done in this manuscript. So, the current study certainly adds information, but the precise value of these new experiments needs to be better worked out. Thus, the current introduction needs serious rewording.

"Our results suggest that the function of the retina and/or the intrinsic photosensitivity of ipRGCs-like-cells of the pineal gland "

> Suggest to re-word to „ Our results suggest that the function of the retina and the intrinsic photosensitivity of ipRGCs-like-cells of the pineal gland expressing opn4xa …”.

Alternatively- refer to the fos stainings and make clear that your data suggest that at these early stages opn4xa appears to be the only opsin mediating light input into the pineal with light of the given duration, specific spectrum and intensity.

There are more opsins expressed in the pineal gland, and hence without further specifications it is difficult to conclude from the lack of one of the opsins about the functionality of the entire cell type.

Results:

>Please add information on the age of the tested fish into Fig1. It’s in the methods, but as this information is important it should be mentioned directly with the results, especially as different ages are used in Fig.3. (And exposing fish for 5 days to LD does not necessarily imply that the test age starts with 5dpf.)

> Before the mansucripts is published it MUST add information on light intensity and spectrum in its supplementary material. This information is absolutely critical when presenting results on photoreceptors.

> Based on the phase entrainment experiments with the lak mutants, the authors conclude that “Retinal Ganglion Cells are differentially required in a phase delay and a phase-advance paradigm of photoentrainment”. This interpretation needs some revisions. The authors cannot really interpret their results as a test for the impact of RGCs. lak mutants are very differently pigmented compared to their wildtype counterparts, which implies that less light will be able to reach all the tissues of the mutant fish compared to wildtype siblings.

Also, where is ath5, the gene mutated in lak mutant fish, expressed besides the eyes? This should be mentioned and considered for the interpretation of the results.

A possible way to discriminate between RGC vs. pigment vs. ath5 impacts on phase advance entrainment would be to repeat these experiments with wildtype vs. lak-/- eye-ablated larvae. However, the reviewer is aware that this will likely require a new animal experimental permit request. No matter the country- these legal permits usually take months to get. Thus, a vigorous discussion of the different interpretational alternatives would be an appropriate approach to address the raised concern, if such a permit does not already exist.

> Another concern on the PA results is the aspect that is really not well visible in the averaged data from the three experiments. The authors address the issue by showing a single experiment in the supplemental, which is really helpful. It would be even more convincing if they include all three experiments individually in the supplemental materials.

> The diel expression changes of opn4xa in interneurons of the eye other than RGC is interesting. A suggestion for improvements: This needs some rewording as RGCs are also interneurons. The anatomy and cell types of the vertebrate eye are very well described. The authors should be able to assign a more specific label to the cells- are these likely amacrine or horizontal or bipolar cells? This should be possible to tentatively assign based on position and cellular morphology. This provides valuable information for future studies.

> The difference in the fos staining is very interesting, as it indeed suggests that much of the pineal light perception is via opn4xa during these larval stages. This is remarkable given many opsins are present in the fish pineal. This could be given some emphasis in the text.

„Similarly to the analysis we performed in lak larvae, for each of these condition“

> Should be „similar“.

Discussion:

"Melanopsin expressing cells of the eye are the sole mediators of light input to the circadian rhythm in

nocturnal mammals."

> ipRGCs are also considered the sole mediators of entrainment by light in humans- i.e. diurnal mammal. Please rephrase the sentence.

>It would be important really interesting to see how the authors embed their work into the current general knowledge on Opsin functions in teleost fish. There are many more and many even described to be expressed in the pineal. In this context the fos expression comparison between mutant and wildtype is particularly remarkable. Do the authors expect that this sole role of opn4xa changes with later stages of development? What’s known about possible alterations of opsin functions in teleosts? Or do they think that this is due to the specific spectrum/intensity of light they use?

>The argument with the “destabilized” clock is cryptic. First, what exactly LL is doing depends very much on its intensity. This information is entirely lacking (see above). Second, moderate LL will cause differences in period length (see Aschoff, J. Cold Spring Harb Symp Quant Biol. 1960; 25:11-28, but besides the mentioned examples that’s also established for humans, the insect Drosophila, the annelid Platynereis). The mentioned “destabilization” likely refers to an effect seen under strong LL when it is assumed that this causes a constant re-setting of the circadian phase and hence desynchronization of the cellular networks and ultimately behavior.

One possible alternative explanation for the observed effects on the LL period of opn4xa mutants is that they sense less light (i.e. less light reaches the fish’s circadian clockwork) and hence– consistent with the above mentioned observations in many organisms– mutants show a slight period length difference.

Reviewer #2: In this study, Chaigne et al. examine the locomotor activities of a zebrafish developmental mutant lakritz (lak), and a CRISPR-Cas9-generated melanopsin gene opn4xa mutant, as well as their responses to two-pulse skeleton photoperiods. The authors reported that retinal ganglion cells and Opn4xa are not required for generating zebrafish locomotor rhythms. They also showed that functional eyes with retinal ganglion cells, but not Opn4xa, contribute to what they called “masking.” These are interesting observations. However, numerous issues must be carefully addressed.

1. Even though the circadian functions of intrinsically photosensitive retinal ganglion cells (ipRGCs) have been most revealed in nocturnal mice. Studies have shown that diurnal rodents, even humans, likely possess functional equivalent ipRGCs as nocturnal mice (doi:10.1371/journal.pone.0073343, doi.org/10.3389/fneur.2021.636330). Thus, in the title, as well as numerous descriptions and discussions in the manuscript, the authors should consider the photoentrainment differences between diurnal zebrafish and nocturnal mice carefully, as these differences are not just between diurnal zebrafish and nocturnal mice.

2. The authors should examine possible mechanisms that underpin what they observed in locomotor activities. For instance, the expression of two light-inducible circadian clock genes, per2 and cry1a(a), in lak and opn4xa mutants should be done.

3. The authors should use rhythmicity programs, such as Hughes et al. 2010. JTK_CYCLE: an efficient nonparametric algorithm for detecting rhythmic components in genome-scale data sets. J. Biological Rhythms) and/or MetaCycle (Wu et al. 2016. MetaCycle: an integrated R package to evaluate periodicity in large scale data. Bioinformatics) to analyze their behavioral data, which produces, rhythmicity probability, period, amplitude, and phase of each data set. For instance, in Figures 1C and E, it may have differences in amplitudes and phases between lak and sib.

4. In Figure 1, why D used “four complete cycles in DD” while D used “three complete cycles in DD”?

5. In the two-pulse skeleton photoperiod experiments, why a two-hour pulse of light at CT 16 while a one-hour pulse of light at CT21? What is the rationale?

6. What machine was used to conduct behavioral assays? Was it homemade?

7. As most of the behavioral assays were done after 6 day post fertilization, when larvae start foraging, were these larvae fed?

8. What is the lux (intensity) of light used?

9. Δphase calculated via the difference of phase between the two last days (“after the light pulse”) and the two first days (“before the light pulse”) may not be accurate because of the developmental stage-specific effects. Δphase should be the phase difference between larvae with the light impulse and larvae without the light impulse of the same period.

10. It would be better to show the data in Tables 1-3 as Figures.

11. In Figure 3, B, C, and D should have scale bars. Quite contrary to what is claimed in the manuscript, I see differences in the numbers of positive (opn4xa and c-fos) cells, as well as their spatial structures between opn4xa mutant and wild-type control larvae.

12. Where is the tcf7 expression data?

13. No figure legend for supplementary figure 3. Scale bars should be shown in A-D.

14. Put “To genotype opn4xa individuals, we used a classical PCR protocol with the following oligos: 5'-

GGACGCCTCCAAACTTC-3' (Forward) and 5'-CGAACACCCACTCCTTGTAC-3' Reverse). PCR products of different sizes were obtained (110bp for the wt allele and 127bp for the mutant allele) and resolved on a 4% agarose gel” into “Generation of an opn4xa mutant allele”.

15. “Enucleated mice” should be “eye enucleated mice”?

16. “Mice mutant for Opn4 show” should be “Mice mutant for Opn4 shows”

17. “In contrast, mice with no ipRGCs or with impaired neurotransmission from ipRGCs show no entrainment to LD as well as no phase delay following a light pulse at CT16 (Cahill, 1996; Güler et al., 2008; Kofuji et al., 2016).” Is Cahill, 1996 relevant here?

Reviewer #3: This manuscript by Chaigne et al provides some interesting new insight into the genetic and molecular elements that contribute to how environmental lighting conditions are detected and thereby entrain the vertebrate circadian clock. The authors focus on this issue using the diurnal, non-mammalian model, the Zebrafish instead of more commonly used nocturnal rodents such as mouse. More specifically, they explore the role of retinal ganglion cells (RGCs) and the zebrafish melanopsin photoreceptor Opn4xa in the entrainment of locomotor activity rhythms of zebrafish larvae. Using the lakritz mutant which lacks RGCs, they reveal attenuation of both the masking effects of light on locomotor activity in larvae as well as the phase advances induced by a pulse of white light delivered at the end of the subjective night period. Instead, the establishment of robust circadian cycles of locomotor activity as well as the phase delaying effects of light pulses delivered at the beginning of the subjective night, are all normal in the lakritz homozygous mutants. The next step is the generation of a loss of function mutation in the Opn4xa gene. Mutant fish exhibit apparently normal masking and circadian rhythms of locomotor activity as well as phase shifting responses to light pulses. The only abnormality detected in these fish are changes in the period length of activity rhythms as they free run in conditions of constant light. The authors conclude that there are „profound“ differences between the diurnal zebrafish and nocturnal rodents such as the mouse in terms of circadian system and photoentrainment.

While certainly interesting, the manuscript in its current form has a number of conceptual and experimental weaknesses as well as problems in the way that the work is presented that preclude its publication in its current form.

Issues, in their order of appearance in the manuscript:

Title, Abstract and Introduction

1) For a number of reasons, I find the title somewhat misleading: The word „profound“ is an extremely strong word and I do not agree it is appropriate for the findings that have been reported here. Also, the reference to diurnal and nocturnal implies that somehow the results have ascribed special significance to the temporal niche occupied by these two species. I don’t see this.

2) First sentence of abstract (as well as first sentence of Results): the importance of the eye for controlling circadian rhythms is not a special feature limited to „nocturnal rodents“ (- the inference here). It is also very clearly the case for other diurnal vertebrates such as humans. This recurrent reference to diurnality and nocturnality throughout the text needs to be toned down.

3) There is already much evidence for major differences between the organization of the zebrafish and mouse circadian timing systems, so the last sentence of the abstract (as well as the title) which implies that this is somehow the original conclusion of this work is also misleading.

4) First sentence of the Introduction: „regulate a phenomenal variety of biological functions“ is once again a wild over statement!

5) End of second paragraph of the Introduction section. „…which is though to behave as a ‚master clock‘….“ is a little understating the issue. There is so much data to support this notion that I think it would be more accurate to state „…..is well established to serve as a master clock”.

6) Introduction, paragraph 4: While the cavefish P. andruzzii does not show significant locomotor activity rhythmicity under LD cycle conditions (Cavallari et al., 2011), Astyanax mexicanus Pachon (blind form) does show rhythmicity under artificial LD cycles (Beale et al., 2013). This statement should be corrected. Last sentence, „peripheric“ should be corrected to „peripheral“, and „………still begs for an answer“ should be corrected to „…. is still unclear“

Results and Conclusions

1) The authors should provide a more detailed description of the Lakritz mutant retinal phenotype as well as details of the underlying mutation in Ath5 etc.

2) 2nd paragraph: „The reduction in activity observed during the day in LD conditions thus does not reflect a defect in circadian control“ is not an accurate statement based on the data that this statement is based on. Better would be „The reduction in activity observed during the day in LD conditions thus does not affect the persistence of rhythmicity under free running conditions in DD“.

3) The opn4xa knock out experiments completely lack formal proof that the mutation indeed does lead to production of a significantly truncated and non-functional protein!!!!

4) Light-induced fos expression in the pineal gland is used as a proxy for „intrinsic photosensitivity in pineal cells“. However, this assumes that there is just a single pathway linking Opn4xa with gene expression. I am not convinced that there is sufficient data to support such a hypothesis. Therefore, it would be important to test expression of other light-regulated genes such as 6-4 photolyase, Cry1a or Per2 in these pineal cells.

5) In the discussion, the authors refer correctly to the large opsin gene repertoire of zebrafish and how this might complicate the interpretation of their results. Given this central issue, it is important that the authors defend in more detail precisely why this particular opsin was chosen for the loss of function experiments. Furthermore, it might be valuable for the reader to already mention this issue in the Introduction..

**Have all data underlying the figures and results presented in the manuscript been provided?**

Reviewer #1: **No: **The authors state that all the numerical data will be sent as supplemental information in the form of excell files with the revised manuscript. This has not been done yet.

Reviewer #2: **No: **No data for tcf7 expression

Reviewer #3: Yes

PLOS authors have the option to publish the peer review history of their article (what does this mean?). If published, this will include your full peer review and any attached files.

Reviewer #1: No

Reviewer #2: No

Reviewer #3: No

---

## [Decision Letter · Decision Letter 1]

12 Jul 2023

Dear Dr Cau,

Thank you very much for submitting your revised Research Article entitled 'Contribution of the eye and of opn4xa function to circadian photoentrainment in the diurnal zebrafish' to PLOS Genetics.

The revised manuscript was fully evaluated at the editorial level and by three independent peer reviewers. The reviewers appreciated the revisions made to the original submission, but consider that some substantial concerns about the current manuscript remain. Based on the reviews, we will not be able to accept this version of the manuscript, but we would be willing to review a much-revised version. We cannot, of course, promise publication at that time.

If you decide to revise the manuscript for further consideration at PLOS Genetics, please aim to resubmit within the next 60 days, unless it will take extra time to address the concerns of the reviewers, in which case we would appreciate an expected resubmission date by email to plosgenetics@plos.org.

We are sorry that we cannot be more positive about your manuscript at this stage. Please do not hesitate to contact us if you have any concerns or questions.

Yours sincerely,

John Ewer

Academic Editor

PLOS Genetics

Gregory Barsh

Editor-in-Chief

PLOS Genetics

Reviewer's Responses to Questions

**Comments to the Authors:**

Reviewer #1: The authors have sufficiently addressed my concerns and comments.

Reviewer #2: I appreciate that the authors have addressed some of my concerns and suggestions. However, my main concerns are not adequately responded to.

1. Regarding the Δphase, I suggested it should be the phase difference between larvae with the light impulse and larvae without the light impulse of the same period. Actually, it is the phase difference between those of the larvae under DD and PD/PA, which would be a better estimate.

2. Regarding the possible mechanisms underlying what you observed in locomotor activities, in particular, why “the eye/RGCs are dispensable for the induction of a phase delay following a pulse of white light at CT 16 but contribute to the induction of a phase advance upon a pulse of white light at CT21.” The parallel examination of the expression of two light-inducible circadian clock genes, per2 and cry1a(a), in lak mutant, should be done, rather than just examining the expression of circadian clock genes under LL conditions, as shown in Figure 5. The hypothesis could be that the expression of key circadian clock genes is altered in these light impulse schedules, which may underlie the behavioral phenotypes.

3. Regarding what is shown in Figures 3B and 3C, the authors should select new representative panels that show similar numbers of opn4xa-positive cells in the mutant and wild-type controls.

4. It is highly recommended that the authors should put some paramecia in each well, along with tested larvae, which should help obtain better behavioral data, avoiding having the data shown in Figure 1 B, where the third-day results show little difference between the mutant and wild types, likely due to the possibility that the larvae ran out of energy without feeding.

5. It is good that the authors performed the rhythmicity analysis of the relevant behavioral data with Biodare2. But the authors should cite the results, for example, locomotor activities of the lak mutant and wild type display rhythmicity with statistically significant P values. Put the P values on the respective panels.

Minor issues:

In Figure 5. Should tefalpha be tefa? What is reverbB1? These genes should have IDs.

Reviewer #3: The revised version of this manuscript is much improved and most of my concerns have been addressed. However, there are some issues which have been raised in the process of revision that I feel need to be addressed before the manuscript can pass on to the final step. Specifically:

1) Text modification 1.111-133

The circadian locomotor activity rhythm phenotype of Astyanax mexicanus is NOT documented in Cavallari et al, 2011. Instead this is refered to in the manuscript from the Whitmore group: Beale et al., 2013, Nature Communications. This citation should be added.

2) Text modification 1.111-133, Final sentence(s): “………..although some level of central control is expected for orchestrating a complex process such as behaviour. As such, the relative importance of central versus peripheral control for the photoentrainment of locomotor activity is still unclear.” Here, and in other parts of the manuscript, the authors tend to give the impression that if a central clock structure is involved in photoentrainment of complex behaviour, then it should operate like the mammalian SCN - so light input coming from the retina or other dedicated photoreceptors. However, as the authors state, the situation in fish is that most cell types, likely including neurons, are directly photosensitive. So, the division between central and peripheral clocks in fish is less well defined. Therefore, it is formally possible that a central CNS-based pacemaker could coordinate light entrainment of rhythmicity WITHOUT the need for indirect input from a functional photoreceptor - in some way acting more like a peripheral clock. This possibility should be raised to avoid any confusion.

3) Text modification 1.161-164

The text should be corrected to “……. is expressed exclusively in the developing retina (Masai et al., 2000). The Lakth241 allele we used bears a point mutation and functions as a null allele which results in a total absence of both…….”

4) Concerning the lack of experimental evidence for the production of a truncated Opn4xa opsin protein.

I apologise that I do not agree with the authors in their argumentation. Using the observation that most zebrafish papers in which a mutant line is generated do not show direct protein data is not a credible excuse. If a mouse-based project failed to provide such evidence then this would be considered as a serious negative point. I am sure that antibodies can be raised against proteins even from zebrafish - it just takes more time and effort!!! However, that said I do appreciate the authors arguments that protein prediction can be useful as indirect evidence that the mutation is likely to have been successful. As a compromise solution, I would advise the authors to state in the Results section text the lines of evidence that they have provided in their rebuttal letter, basically that they have eliminated the possibility of Alternative ORFs, Alternative ATGs and alternative splicing. At least this way, the reader can make up their own mind as to the confidence they can associate with the results obtained using this mutant.

5) The logic for requesting expression data for Cry1a, and 6-4 photolyase: These (as well as per2) have been extensively studied as light regulated genes, and the enhancers responsible for this photic regulation have been identified. There are also gene-specific differences in the mechanisms (TFs) whereby they are regulated by light. Furthermore, given the complexity of the opsin gene repertoire in the zebrafish, it seems likely that there is also complexity at the signalling and gene expression regulation levels of the light response. For this reason, analysing a small number of well characterized light - regulated genes would provide more compelling evidence for whether a particular cell type is truly photosensitive or not. However, on balance, I do not hold the authors to provide this extra evidence.

**Have all data underlying the figures and results presented in the manuscript been provided?**

Reviewer #1: Yes

Reviewer #2: Yes

Reviewer #3: Yes

PLOS authors have the option to publish the peer review history of their article (what does this mean?). If published, this will include your full peer review and any attached files.

Reviewer #1: No

Reviewer #2: No

Reviewer #3: No

---

## [Decision Letter · Decision Letter 2]

8 Jan 2024

Dear Dr Cau,

Thank you very much for submitting your re-revised Research Article entitled 'Contribution of the eye and of opn4xa function to circadian photoentrainment in the diurnal zebrafish' to PLOS Genetics.

The manuscript was fully evaluated at the editorial level and by Reviewer 2. You will see that Reviewer 2 still has reservations about the work. Although we agree with their comments, the Editor-in -Chief and I agree that they do not severely jeopardize this solid piece of work. Nevertheless, we ask you to please respond to the reviewer's comments as best you can. And in order to not extend this reviewing process any further, the revised version will not be sent out for review again; it (together with your detailed response to the reviewer's comments) will be evaluated at the editorial level only.

Thus, we ask you to please:

Yours sincerely,

John Ewer

Academic Editor

PLOS Genetics

Gregory Barsh

Editor-in-Chief

PLOS Genetics

Reviewer's Responses to Questions

**Comments to the Authors:**

Reviewer #2: This reviewer appreciates that the authors took time to address the issues being raised. However, I felt that several issues are still not adequately explained.

1. Regarding the Δphase, I think that the difference between the final phase (‘phase after’) and the initial phase (‘phase before’) of the same larvae doesn’t faithfully reflect the effects of light impulses on phase changes of locomotor activities, even it might get a better value, as the authors argued. As shown in Figures 2D and 2E, compared to those in DD, light impulses at CT16 lead to phase delay, while light impulses at CT 21 cause phase advance. Thus Δphase between those of PD larvae and those of DD control larvae, and between those of PA larvae and those of DD control larvae should be a better indicator of the effects of light impulses on phase changes. With any behavioral assay, interindividual and interexperimental variations often occur, which should not be a reason not to calculate the right parameter. If the authors have calculated Δphase between those of PD larvae and those of DD control larvae, and between those of PA larvae and those of DD control larvae, they should describe them in the main text. I think that Δphase between the final phase (‘phase after’) and the initial phase (‘phase before’) of the same larvae is just arrow phase change, which will confuse the audience, should be deleted or put in to the supplementary file.

2. The authors said that “we addressed whether per2 and cry1a were induced upon administration of a one-hour light pulse at CT21 and how these inductions were affected by the absence of a functional lak gene. These results are described in figure 5”, I did not see them in Figure 5.

3. The authors cited studies that paramecia display circadian rhythms, and argued that paramecia feeding could affect the larval locomotor rhythms. The fact is that the authors showed the amplitude damped quickly in their behavioral assays without feeding, while many behavioral experiments display robust behavioral rhythms with paramecia feeding.

4. I suggest the authors should cite the results of Biodare2, for example, locomotor activities of the

lak mutant and wild type display rhythmicity with statistically significant P values. Put the P

values on the respective panels. The author didnot give reasons why not putting the P

values on the respective panels.

5. In Figure 5, tefalpha is still not changed into tefa.

**Have all data underlying the figures and results presented in the manuscript been provided?**

Reviewer #2: Yes

PLOS authors have the option to publish the peer review history of their article (what does this mean?). If published, this will include your full peer review and any attached files.

Reviewer #2: No

---

## [Editor Report · Decision Letter 3]

5 Feb 2024

Dear Dr Cau,

We are pleased to inform you that your manuscript entitled "Contribution of the eye and of opn4xa function to circadian photoentrainment in the diurnal zebrafish" has (finally!) been editorially accepted for publication in PLOS Genetics. Congratulations! I made a couple of suggestions for the text (see below) but you should be able to make them in the proofs, if not before.

Yours sincerely,

John Ewer

Academic Editor

PLOS Genetics

Gregory Barsh

Editor-in-Chief

PLOS Genetics

Comments from the reviewers (if applicable):

Thank you for making the final changes to the manuscript. My only suggestions would be to change "... supporting that vertebrates might employ..." (lines 41-42) for ".... revealing that vertebrates employ...". I think your results are solid enough to make a more definitive statement. Also line 226 needs editing. I would suggest "... it is more apparent when comparing only the records of animals for which a phase was...." You should be able to request these small changes when you are sent the proofs.

**Data Deposition**

http://datadryad.org/submit?journalID=pgenetics&manu=PGENETICS-D-22-01386R3

**Press Queries**

---

## [Editor Report · Acceptance letter]

21 Feb 2024

PGENETICS-D-22-01386R3 

Contribution of the eye and of *opn4xa* function to circadian photoentrainment in the diurnal zebrafish 

Dear Dr Cau, 

We are pleased to inform you that your manuscript entitled "Contribution of the eye and of *opn4xa* function to circadian photoentrainment in the diurnal zebrafish" has been formally accepted for publication in PLOS Genetics! Your manuscript is now with our production department and you will be notified of the publication date in due course.

With kind regards,

Anita Estes

PLOS Genetics

On behalf of:
